# A spool for every quotient:
# One-loop partition functions in AdS$_3$ gravity

Robert Bourne, Jackson R. Fliss, Bob Knighton

*Department of Applied Mathematics and Theoretical Physics, University of Cambridge,*
*Cambridge CB3 0WA, United Kingdom*

`rjb260@cam.ac.uk, jf768@cam.ac.uk, rik23@cam.ac.uk`

## Abstract

The Wilson spool is a prescription for expressing one-loop determinants as topological line operators in three-dimensional gravity. We extend this program to describe massive spinning fields on all smooth, cusp-free, solutions of Euclidean gravity with a negative cosmological constant. Our prescription makes use of the expression of such solutions as a quotients of hyperbolic space. The result is a gauge-invariant topological operator, which can be promoted to an off-shell operator in the gravitational path integral about a given saddle-point. When evaluated on-shell, the Wilson spool reproduces and extends the known results of one-loop determinants on hyperbolic quotients. We motivate our construction of the Wilson spool from multiple perspectives: the Selberg trace formula, worldline quantum mechanics, and the quasinormal mode method.

# 1 Introduction

Quantum gauge theories display interesting interplays between local and non-local physics. Their expression as fields interacting locally comes at the expense of introducing non-physical redundancies. Two extreme examples of the tension between locality and gauge invariance are topological field theories – in which *all* local degrees of freedom are redundant and gauge invariant observables are extended and insensitive to geometry – and general relativity – in which the redundancy lies in the local coordinate frame itself. In two and three spacetime dimensions these examples intersect and gravity itself can be described as a topological field theory [1–4].

Matter couples locally to the metric which potentially spoils the topological nature of low dimensional gravity. However gauge invariance requires that local operators be dressed to boundaries or fixed features of a state which turns them, effectively, into extended operators. Remarkably, it was shown in [5–8] that the path-integration over massive matter can result in an effective, topological line operator. More specifically the one-loop determinant of a massive, minimally coupled, field is expressed as an integral of a Wilson loop of the one-form connection(s) encoding the frame and the spin-connection of the background metric. In the examples of [5–8] this Wilson loop wraps a single non-contractible cycle of the background topology and its integration results in a winding around this cycle arbitrarily many times. This operator was coined the *Wilson spool*

apropos of its winding behavior. Specifically for three-dimensional gravity with a negative cosmological constant the Wilson spool on the BTZ background winds around the black hole horizon. For positive cosmological constant and the $S^3$ topology, the spool winds around a defect encoding the location of cosmological horizon after Wick rotation to the static patch of de Sitter.

Due to the relatively simple topology of the above examples, the spool wraps a single non-contractible cycle of the background manifold. However gravity is a theory of geometry and topology, and three-dimensional topology is rich; it includes manifolds with larger and more intricate fundamental groups. This raises the question of how to treat the Wilson spool on manifolds whose fundamental group is generated by more than one element. The treatment of one-loop determinants through worldline quantum mechanics instructs us to sum over all paths in the Euclidean geometry which strongly suggests that the Wilson spool for a given topology should involve a sum over all elements of that topology's fundamental group. Making this physically intuitive notion precise (especially given that a fundamental group will typically involving non-commuting elements with intricate relations) is the primary aim of this paper.

More specifically we will consider all smooth cusp-free hyperbolic three-manifolds. This includes hyperbolic manifolds asymptoting to higher genus surfaces whose Lorentzian interpretations are the asymptotically AdS$_3$ 'multi-boundary wormholes' [9–13], as well compact hyperbolic manifolds obtained from surgery on link complements and which are not (conventionally) holographic; see Figure 1 for examples.

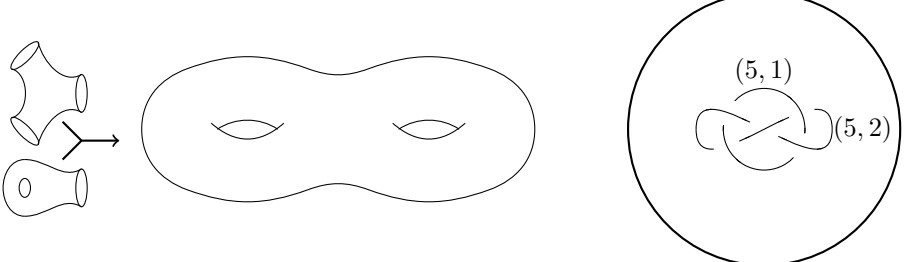

**Figure 1:** Example Euclidean hyperbolic manifolds to which our prescription applies. **(Left)** A solid handlebody whose asymptotic boundary is the genus two Riemann surface. Depending on the detail of the quotient, this handlebody arises as Euclidean rotation of the Lorentzian spacetime whose time-symmetric slice has three asymptotic boundaries, as well as a Lorentzian spacetime with a single asymptotic boundary and a torus hidden 'behind the horizon' [11]. **(Right)** The Weeks manifold, the compact hyperbolic manifold of smallest volume, obtained by the labeled Dehn surgeries on the Whitehead link embedded in $S^3$.

These are all smooth vacuum saddles of Euclidean Einstein-Hilbert gravity with a negative cosmological constant. As we will explain in more detail below these manifolds can be expressed as quotients of the hyperbolic three-ball $\mathbb{H}^3$ by the action of a class of discrete subgroup, $\Gamma$, of PSL$(2, \mathbb{C})$, the isometry group of $\mathbb{H}^3$:

$$M = \mathbb{H}^3/\Gamma \ . \tag{1.1}$$

We will refer to such three-manifolds as *smooth hyperbolic quotients.* The subgroup, $\Gamma$, is identified in a natural way with the fundamental group of $M$ and the conjugacy class, $[\gamma]$, of any $\gamma \in \Gamma$ can be identified with a free loop in $M$. We will give a prescription for coupling massive scalar and spinning fields to such backgrounds in a way that makes the topological nature of three-dimensional gravity manifest and utilizes directly the manifold's fundamental group. To be specific, given the local path-integral, $Z_{\Delta,\mathsf{s}}$, of a massive spin-$\mathsf{s}$ field (described by a transverse, traceless, and symmetric $\mathsf{s}$-tensor along with a tower of associated Stückelberg fields) minimally coupled to the metric, $g_{\mu\nu}$, of a manifold, $M$, that is diffeomorphic to the hyperbolic quotient $\mathbb{H}^3/\Gamma$, we express

$$\log Z_{\Delta,\mathsf{s}}[g_{\mu\nu}] = \mathbb{W}_\Gamma[A_L, A_R] \ , \tag{1.2}$$

where

$$\mathbb{W}_\Gamma = \sum_{[\gamma]_+} \sum_{\mathsf{R}_{L/R}} \frac{1}{n_\gamma} \left[ \mathrm{Tr}_{\mathsf{R}_L} \mathcal{P} \exp \left( \oint_\gamma A_L \right) \right] \left[ \mathrm{Tr}_{\mathsf{R}_R} \mathcal{P} \exp \left( - \oint_\gamma A_R \right) \right] \ . \tag{1.3}$$

This is the Wilson spool generalized to smooth hyperbolic three-manifolds and is the primary result of this paper. The body of this paper, Section 2, will establish and uphold this result, however let us now point out some broad features of (1.2) and (1.3).

The geometry of the metric, $g_{\mu\nu}$, is encoded in the one-form connections $A_{L/R}$ which are linear combinations of the coframe and spin-connection of $g_{\mu\nu}$. The details of this well-known map are reviewed in Section 2.1. The representations $\mathsf{R}_{L/R}$ encode the mass and spin of field and (1.3) instructs us to sum over the representations possessing same sum of quadratic Casimirs, $c_2^{\mathfrak{sl}(2,\mathbb{R})_L} + c_2^{\mathfrak{sl}(2,\mathbb{R})_R}$, which we explain in Section 2.3. The one-form connections are integrated over an oriented free-loop, $\gamma$, of $M$ which then corresponds to a unique conjugacy class of $\Gamma$; above we choose the class $[\gamma]_+$, resulting in a strictly positive geodesic lengths. The spool then sums over all such non-trivial conjugacy classes weighted by the inverse of their *multiplicity*, $n_\gamma$. We will explain in Section 2.2 how this factor algebraically accounts for the relations of the fundamental group of $M$. In Section 3.2.2 we will cast it in a much more intuitive role as a symmetry factor upon moving to the $\mathbb{H}^3$ cover of $M$ and realizing the Wilson loops in (1.3) as worldline quantum mechanics. It should be clear that in both contexts that $[\gamma]_+$ and $n_\gamma$ are features of the topology of $M$ as opposed to its geometry. The properties of the fundamental group and the properties of the holonomies of the geometric connections, $A_{L/R}$, further allow us to express $\mathbb{W}_\Gamma$ in an integral representation as

$$\mathbb{W}_\Gamma = \frac{i}{2} \sum_{[\gamma_0]_+} \sum_{\mathsf{R}_{L/R}} \int_{\mathcal{C}} \frac{d\alpha}{\alpha} \frac{\cos \alpha/2}{\sin \alpha/2} \left[ \mathrm{Tr}_{\mathsf{R}_L} \mathcal{P} \exp \left( \frac{\alpha}{2\pi} \oint_{\gamma_0} A_L \right) \right] \left[ \mathrm{Tr}_{\mathsf{R}_R} \mathcal{P} \exp \left( - \frac{\alpha}{2\pi} \oint_{\gamma_0} A_R \right) \right] , \tag{1.4}$$

where $\mathcal{C}$ is a curve wrapping tightly clockwise the real $\alpha$ axis and the sum now ranges over conjugacy classes of the *primitive generators*, $\gamma_0$, of $\Gamma$ with strictly positive geodesic length. While this follows simply from counting residues, (1.4) makes clear that when $\Gamma$ possesses a single primitive generator our result reproduces the examples in [6,7].

We will show in Section 3.1 that (1.2) and (1.3) taken on-shell reproduce the one-loop determinants of massive scalar and vector fields, established by [14]; for massive fields of spin $\mathsf{s} \geq 2$, our result provides an expression for their one-loop determinant on any smooth hyperbolic quotient, a result that, to our knowledge, has not appeared in the literature. In Section 3.2, we will uphold (1.2) and (1.3) through three separate 'derivations.' The first of these, in Section 3.2.1, will follow the Selberg trace formula which relates the spectrum of hyperbolic Laplacians and the spectrum of closed geodesics. Secondly, we will relate the one-loop determinant to line operators through worldline quantum mechanics in Section 3.2.2; two key outcomes of this section are that (i) the worldline path-integral is two-loop exact and reproduces the representation characters that appear in (1.3), and (ii) a topological and intuitive interpretation of the multiplicity, $n_\gamma$, as the symmetry factor of a worldline uplifted to the cover of $M$. Lastly we will give a quasinormal mode perspective to (1.2) and (1.3), explaining how $\exp \mathbb{W}_\Gamma$ reproduces the structure of poles of the one-loop determinant, $Z_{\Delta,\mathsf{s}}$. This derivation more closely matches the original derivations of the Wilson spool and we will lean heavily on the technology established in [7]. While these derivations will technically be established for on-shell connections, the resulting operator (1.3) will be expressed completely in terms of topological and gauge-invariant quantities and we will posit that (1.2) holds *off-shell* in a weak sense, i.e. within expectation values of diffeomorphism invariant operators inside the gravitational path-integral.

Finally we conclude this paper with a discussion of our result speculating on more general hyperbolic quotients including orbifolds and cusps, and how the Wilson spool fits into the context of recent progress in three-dimensional quantum gravity. Further details on representation theory and hyperbolic quotients used in this paper can be found in the Appendices.

**NB**: As this work neared completion we learned of upcoming work, [15], which has overlap with our results. We have coordinated submissions with the authors of that work.

## 2 Background

In this section of the paper we will build up the necessary frameworks for our result (1.3). We will establish the context in which it is situated by reviewing some basics of the Chern-Simons formulation of three-dimensional gravity, linking topological features of geodesics to holonomies of the Chern-Simons connections, and afterwards review the representation theory of minimally coupled massive matter.

### 2.1 Chern-Simons gravity

We consider Euclidean three-dimensional gravity on locally asymptotically AdS$_3$ manifolds. For our purposes, it will be useful to first arrive at this Euclidean theory from a Wick rotation of the Lorentzian theory. As mentioned in the introduction, the Einstein-Hilbert action in three dimensions can be expressed as a topological field theory and in

Lorentzian signature this takes the form of a pair of Chern-Simons actions for connections $A_{L/R}$ taking values in $\mathfrak{sl}(2,\mathbb{R})_L \oplus \mathfrak{sl}(2,\mathbb{R})_R$:

$$S_{\mathrm{EH}} = k\, S_{\mathrm{CS}}[A_L] - k\, S_{\mathrm{CS}}[A_R] \ , \qquad S_{\mathrm{CS}}[A] = \frac{1}{2\pi} \int_{M_3} \mathrm{Tr}\left( A \wedge \mathrm{d}A + \frac{2}{3} A^3 \right) \ , \qquad (2.1)$$

where the Tr is taken in the fundamental representation. This rewriting is facilitated by the identification

$$A_L = (\omega^a + e^a/\ell) L_a \ , \qquad A_R = (\omega^a - e^a/\ell) \bar{L}_a \ , \qquad (2.2)$$

where $\{L_a\}$ and $\{\bar{L}_a\}$ generate $\mathfrak{sl}(2,\mathbb{R})_L$ and $\mathfrak{sl}(2,\mathbb{R})_R$, respectively,[1] $e^a$ are the coframes, $\omega^a$ are the dual spin-connections, and $\ell$ is the AdS radius. The Chern-Simons level is related to Newton's constant via

$$k = \frac{\ell}{4G_N} \ . \qquad (2.3)$$

We now perform the Wick rotation to Euclidean signature through

$$e^0 \to -i e^0 \ , \qquad L_0 \to i L_0 \ , \qquad \omega^{1,2} \to i\omega^{1,2} \ , \qquad (2.4)$$

to write

$$A_L = (i\omega^a + e^a/\ell)\, L_a \ , \qquad A_R = (i\omega^a - e^a/\ell)\, \bar{L}_a \ . \qquad (2.5)$$

The Euclidean isometry algebra, $\mathfrak{so}(1,3)$, does not split, however we regard $A_{L/R}$ as components of a real form in $\mathfrak{sl}(2,\mathbb{C})$ which is the common complexification of $\mathfrak{so}(1,3)$ and $\mathfrak{so}(2,2)$.

While pure 3D gravity with negative cosmological constant is classically equivalent to $\mathrm{SL}(2,\mathbb{R})_L \times \mathrm{SL}(2,\mathbb{R})_R$ Chern-Simons theory at the level of the action, it is important to emphasize that they are not fully equivalent as quantum theories. The most obvious reason for this is that in gravity one should only integrate over metrics which are invertible, while no such restriction is imposed in Chern-Simons theory – indeed, the trivial connection $A_L = A_R = 0$ is a perfectly valid gauge configuration corresponding to an everywhere-vanishing metric. Another difference is that in gravity one in principle should sum over all bulk geometries – including those of differing topologies – consistent with the boundary conditions of the problem, whereas the path-integral of a topological gauge theory is usually seen as an assignment of a complex number to a fixed topology. These issues can be addressed by isolating the proper moduli space of Chern-Simons connections corresponding to invertible metrics [16, 17] and summing over topologies by hand. However other issues (such as a continuous and sometimes negative density of states [18]) require more effort to address [19]. In this sense, the Chern-Simons description should be seen as an effective (albeit UV finite) theory of pure three-dimensional gravity. In this paper, we will be interested in computing one-loop determinants of matter fields around fixed hyperbolic metrics in this effective description. Thus, while important, the differences between pure 3D gravity and Chern-Simons theory do not play a role in our work.

---

[1]Our algebra conventions are given in Appendix A.

### 2.1.1 Linking geometry and holonomy

In constructing the Wilson spool it will be important to understand how to relate geometric properties of closed curves to properties of gauge-invariant operators in the Chern-Simons formulation. Let $M$ be a smooth oriented three-manifold equipped with a fiducial background metric $g_{\mu\nu}$ which may or may not be on-shell. Consider a closed geodesic, $\gamma$, embedded within $M$, and parameterized as $\gamma = \{x^\mu(s)\}$ for an arclength $s$. To $\gamma$ we can associate two natural coordinate invariant quantities: its geodesic *length* and its *torsion*, which we define in the following way.

Denote the unit tangent vector to $\gamma$ as $\mathsf{t}$ and its dual as $\mathrm{d}\mathsf{t}$.[2] The geodesic length, $l_\gamma$, then is the integral

$$l_\gamma = \int_\gamma \mathrm{d}\mathsf{t} \ . \tag{2.7}$$

The torsion, $\theta_\gamma$, arises in the following way. We first define the local extrinsic torsion as

$$\vartheta = \frac{1}{2}\epsilon_{AB}\, n_\mu^A \nabla_{\mathsf{t}} n^{\mu B}\mathrm{d}\mathsf{t} \ , \tag{2.8}$$

where $\{n_A\}_{A=1,2}$ is a basis of the normal bundle. $\vartheta$ behaves as a one-form along $\gamma$, however transforms as a connection in the normal indices. That is, under rotations of the local normal frame by an angle $\psi(s)$, $\tau$ transforms as

$$\vartheta \to \vartheta + \partial_s \psi \, \mathrm{d}s \ . \tag{2.9}$$

The integral of the local extrinsic torsion is the torsion

$$\theta_\gamma = \int_\gamma \vartheta \ . \tag{2.10}$$

and measures the failure of the normal frame to return to itself under parallel transport along $\gamma$.[3] See [20] for a recent review.

Associated with $\gamma$ is a set of Fermi normal coordinates which we construct in the following way. Let $\{\tilde{e}_a\}_{a=0,1,2}$ be an orthonormal frame at $\gamma(0)$ such that $-\tilde{e}_0 = \mathsf{t}$.[4] Parallel transporting this basis along $\gamma$ extends this an orthonormal frame along the whole geodesics such that $\tilde{e}_0(s)$ remains tangent to the curve. At any given $s$, we can shoot

---

[2]Explicitly, in terms of the local parameterization,

$$\mathsf{t}^\mu = \left(\sqrt{g_{\mu\nu}\dot{x}^\mu\dot{x}^\nu}\right)^{-1}\dot{x}^\mu \ , \qquad \mathrm{d}\mathsf{t} = \sqrt{g_{\mu\nu}\dot{x}^\mu\dot{x}^\nu}\,\mathrm{d}s \ . \tag{2.6}$$

[3]Consider a closed curve in general dimension $d$. A normal vector $n$ at some point on the curve, and the same vector parallel transported once around the curve are related by an element of $SO(d-1)$, the structure group of the normal bundle. This is the holonomy of the curve, which in $d=3$ reduces to an angle $\theta \in [0, 2\pi)$

[4]The sign is obviously a convention and we choose this convention to ultimately align more closely with previous constructions [5,7] and standard literature [14].

geodesics normal to $\gamma$ with initial velocity given by $v = -v^1\tilde{e}_1 + v^2\tilde{e}_2$. This establishes a coordinate chart, $(s, v^1, v^2)$, in a tubular neighborhood of $\gamma$ with vanishing Christoffel symbols on $\gamma$ itself. However, as we established above, under parallel transport $\{\tilde{e}_a\}$ may fail to return to itself when traversing $\gamma$ and $\theta_\gamma$ measures this failure. We can amend this by building an alternative frame along $\gamma$, $\{e_a(s)\}$, such that $e_0(s) = \tilde{e}_0(s)$, and with $e_{1,2}$ gradually rotated with respect to $\tilde{e}_{1,2}$ such that they remain periodic along $\gamma$. See Figure 2 for a cartoon.

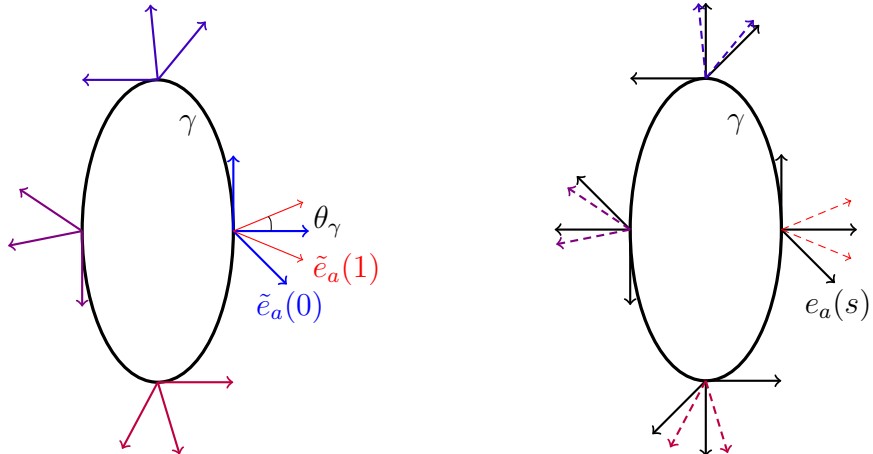

**Figure 2: (Left)** The frame $\tilde{e}_a(s)$ picks up a rotation of $\theta_\gamma$ under parallel transport along $\gamma$. **(Right)** We define a new frame, $e_a(s)$, which is gradually rotated from $\tilde{e}_a(s)$ to undo this holonomy. While $e_a(s)$ is not parallel transported, it does remain periodic along $\gamma$.

This frame maintains that $-e_0(s) = \mathsf{t}(s)$,[5] and satisfies the geodesic equation of motion:

$$e_0^\nu \nabla_\nu e_0^\mu = 0 \ . \tag{2.11}$$

Treating $T\gamma$ as a subspace of $TM|_\gamma$, the coframe, $e^0$, acts as the unit one-form along the curve,

$$-\gamma^* e^0 = \mathsf{dt} \ , \tag{2.12}$$

while

$$\gamma^* e^1 = \gamma^* e^2 = 0 \ . \tag{2.13}$$

The associated dual spin connection

$$\omega_\mu^a = \frac{1}{2}\varepsilon^a{}_{bc}\, e_\nu^b \nabla_\mu e^{\nu c} \tag{2.14}$$

satisfies

$$-\gamma^* \omega^0 = -e_\mu^1 e_0^\nu \nabla_\nu e^{\mu 2}\, e^0 = \vartheta \ , \tag{2.15}$$

---

[5]Note that keeping both the $\{e_a(s)\}$ and $\{\mathsf{t}(s), n_A(s)\}$ frames right-handed also requires us to take $e_1(s) = -n_1(s), e_2(s) = n_2(s)$, a reversal of orientation within the 2d normal bundle.

while (2.11) implies
$$\gamma^*\omega^1 = \gamma^*\omega^2 = 0 \ . \tag{2.16}$$

Notice that this final statement is the only one which relies on $\gamma$ to be geodesic. Because we have not made any assumptions about the background metric being on-shell, we can relax this assumption to any $\gamma$ that is a smooth closed curve homotopic to a geodesic. Then there exists a metric for which $\gamma$ is geodesic and we can define $l_\gamma$ and $\theta_\gamma$ with respect to that metric.[6] It will be convenient to define a *complex length* which captures both the length and the torsion as
$$\hat{l}_\gamma \equiv l_\gamma + i\theta_\gamma \ , \tag{2.17}$$

and an associated nome, $q_\gamma$, and modulus, $\tau_\gamma$, as

$$q_\gamma \equiv \exp\left(-\hat{l}_\gamma\right) \equiv \exp\left(2\pi i \tau_\gamma\right) \ . \tag{2.18}$$

We now consider how the above is expressed in terms of Chern-Simons quantities. The gauge invariant operators of Chern-Simons theory are Wilson loops. Given some representation $\mathsf{R}$ of $\mathfrak{sl}(2,\mathbb{R})$ and a closed oriented path $\gamma : S^1 \to M$ we can construct

$$\mathrm{Tr}_\mathsf{R}\,\mathcal{P}\exp\oint_\gamma \gamma^* A_{L/R} \ , \tag{2.19}$$

where $\gamma^* A_{L/R}$ is the pullback of the connections onto the curve,[7] and $\mathcal{P}$ is the path ordering consistent with the orientation of $\gamma$. These objects are gauge-invariant and we are free to work in any gauge defined locally along the geodesic. In particular, the coframe adapted to $\gamma$ that we described above corresponds to a particular choice of gauge in the Chern-Simons description of gravity. Within this gauge/coordinate frame we then write

$$\mathrm{Tr}_\mathsf{R}\,\mathcal{P}\exp\oint_\gamma \gamma^* A_L = \mathrm{Tr}_\mathsf{R}\,\mathcal{P}\exp\oint_\gamma (i\gamma^*\omega^a + \gamma^* e^a) L_a \tag{2.20}$$

$$= \mathrm{Tr}_\mathsf{R}\,\mathcal{P}\exp\oint_\gamma (-i\vartheta - \mathrm{dt}) L_0 \tag{2.21}$$

$$= \mathrm{Tr}_\mathsf{R}\,\exp\left(-\hat{l}_\gamma L_0\right) \ . \tag{2.22}$$

Similar manipulations show

$$\mathrm{Tr}_\mathsf{R}\,\mathcal{P}\exp\oint_\gamma \gamma^* A_R = \mathrm{Tr}_\mathsf{R}\,\exp\left(\hat{l}^*_\gamma \bar{L}_0\right) \ . \tag{2.23}$$

These expressions are gauge/coordinate independent, and hence hold regardless of our convenient choice of adapted coordinates. The upshot of the above that is we have shown

---

[6]Namely, if $\Phi(\gamma)$ is geodesic with respect to the metric $g_{\mu\nu}$ for some diffeomorphism, $\Phi$, then $\gamma$ is geodesic for the metric $(\Phi^{-1})^* g_{\mu\nu}$.

[7]In the rest of this paper the pullback will be left implicit; we emphasize it here as it plays an important role in the construction.

that the holonomies of the on-shell connections around a cycle, $\gamma$, are given precisely by complex lengths of that cycle:

$$\mathcal{P}\exp\left(\oint_\gamma \gamma^* A_L\right) \sim q_\gamma^{L_0} \,, \qquad \mathcal{P}\exp\left(-\oint_\gamma \gamma^* A_R\right) \sim \bar{q}_\gamma^{\bar{L}_0} \tag{2.24}$$

with ' $\sim$ ' indicating equality up to conjugation and $\bar{q}_\gamma = \exp\hat{l}_\gamma^*$.

At this point we must address the subtlety of orientation. As Wilson loops are oriented observables, each curve $\gamma$ comes with an assigned inherent orientation which, upon embedding into $M$, induces a right-handed orientation in a tubular neighborhood containing it. This orientation may match the ambient orientation of $M$ (or equivalently that of the fiducial coframes associated to $g_{\mu\nu}$) or it may be opposite. However since $\mathrm{SL}(2,\mathbb{R})_L \times \mathrm{SL}(2,\mathbb{R})_R$ gauge symmetry does not contain maps sending $q_\gamma \to q_\gamma^{-1}$,[8] in writing (2.24), we are taking, by convention, the coframes defining (2.12) and (2.15) to match the ambient orientation of $M$. In particular, in this convention, we are allowing $l_\gamma$ to be negative when the intrinsic orientation of $\gamma$ is opposite that of the ambient orientation. We can summarize this as

$$\begin{aligned} |q_\gamma| < 1 \,, \quad &\text{orientation of } \gamma = \text{orientation of } M \,, \\ |q_\gamma| > 1 \,, \quad &\text{orientation of } \gamma = -\text{orientation of } M \,. \end{aligned} \tag{2.25}$$

## 2.2 Hyperbolic quotients

Now let us discuss on-shell contributions to the gravitational path integral which are three-manifolds $M$ admitting hyperbolic metrics. In this paper, we will only focus on smooth, cusp-free, manifolds. As mentioned in Section 1, any hyperbolic three-manifold can be expressed as a quotient

$$M = \mathbb{H}^3/\Gamma \tag{2.26}$$

of global hyperbolic three-space $\mathbb{H}^3$ by a discrete subgroup $\Gamma \subset \mathrm{PSL}(2,\mathbb{C})$, where $\mathrm{PSL}(2,\mathbb{C})$ acts as the group of isometries on $\mathbb{H}^3$. Such a discrete subgroup of $\mathrm{PSL}(2,\mathbb{C})$ is known as a Kleinian group. In this way, the study of hyperbolic 3-manifolds is equivalent to the study of Kleinian groups, and any quantity of interest that can be computed on $M$ can be related to the structure of $\Gamma$. We review this construction in more detail Appendix B, highlighting the necessary features here.

In particular, since $\mathbb{H}^3$ is simply connected, the quotient structure (2.26) implies that the fundamental group of $M$ can be identified with the subgroup $\Gamma$:

$$\pi_1(M, p) \cong \Gamma \,. \tag{2.27}$$

Concretely, this means that every element $\gamma \in \Gamma$ can be associated with a nontrivial loop in $M$ which starts and ends at some fiducial base point $p$. If we do not care about the

---

[8]The group element doing so, $i\sigma_1$ (in the basis diagonalizing $q_\gamma^{L_0}$ and in the fundamental representation), is not an element of either $\mathrm{SL}(2,\mathbb{R})$ however *is* an element of $\mathrm{PSL}(2,\mathbb{C})$.

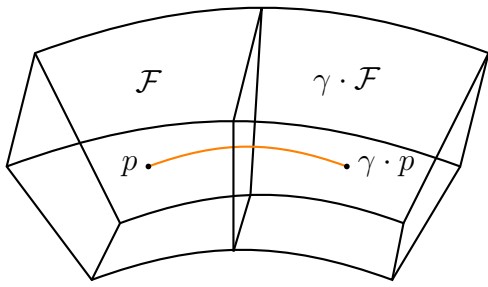

**Figure 3:** Every element $\gamma \in \Gamma$ induces a non-contractable loop $\gamma \in \pi_1(M, p)$ in the quotient space $M = \mathbb{H}^3/\Gamma$.

base point, $p$, the conjugacy class, $[\gamma]$, is associated to a free loop in $M$ up to homotopy. Equivalently, since $M$ is hyperbolic, each conjugacy class $[\gamma]$ is associated to the homotopy class of a closed geodesic in $M$, where a geodesic is constructed with respect to the hyperbolic metric on $M$.

Group elements of $\mathrm{PSL}(2, \mathbb{C})$ are of four different types depending on the value of their trace in the fundamental representation. When $\gamma$ is either *hyperbolic* or *loxodromic* then it is conjugate to

$$\gamma \sim \begin{pmatrix} q_\gamma^{1/2} & 0 \\ 0 & q_\gamma^{-1/2} \end{pmatrix} \ , \tag{2.28}$$

for some $q_\gamma$ with $|q_\gamma| \neq 1$. Such elements act freely on $\mathbb{H}^3$ and fix a single geodesic (as a set). The quotient $M$ is smooth if and only if $\Gamma$ only contains hyperbolic and loxodromic elements. In this case we say that $\Gamma$ is *torsion-free* and in this paper, by focusing on smooth $M$ we will only consider torsion-free $\Gamma$'s.

Within $\Gamma$ there exists a set of primitive elements, $\{\gamma_0\}$, that cannot be expressed as $\gamma_0 = (\gamma_0')^n$ for any $\gamma_0' \in \Gamma$ and $n \geq 1$. Furthermore when $\Gamma$ is comprised solely of hyperbolic and loxodromic elements then *any* $\gamma$ can be expressed as

$$\gamma = (\gamma_0)^{n_\gamma} \ , \tag{2.29}$$

for a primitive element $\gamma_0$ and a power $n_\gamma$ which we call the *multiplicity*. The primitive, $\gamma_0$, is unique up to conjugation and so $n_\gamma$ is an invariant of the conjugacy class of $\gamma$. See Appendix B for further details.

One useful way of viewing the relation between elements of $\Gamma$ and loops in $M$ (which we will use specifically in Section 3.2.2) is by choosing a fundamental domain $\mathcal{F} \subset \mathbb{H}^3$ of the action of $\Gamma$. Given a point $p$ in this fundamental domain, the point $\gamma \cdot p$ lives in another copy $\gamma \cdot \mathcal{F}$ of the fundamental domain, and both $p$ and $\gamma \cdot p$ represent the same point in $M$. Thus, any path in $\mathbb{H}^3$ from $p$ to $\gamma \cdot p$ represents a loop in $M$; see Figure 3.

The covering space formalism described above will prove useful for calculations later in this paper and in this context we can think of the gauge fields $A_L, A_R$ on $M$ as being gauge fields on the covering space $\mathbb{H}^3$ with the restriction that they are periodic with respect to

$\Gamma$ (so that they are single-valued on $M$).[9] In this description, the gauge symmetries are

$$A_{L/R} \rightarrow U_{L/R}^{-1} A_{L/R}(p) U_{L/R} + U_{L/R}^{-1} \mathrm{d} U_{L/R} \, , \tag{2.30}$$

where $U_{L/R}$ are local $\mathrm{SL}(2,\mathbb{R})$ matrices which are periodic with respect to $\Gamma$. Given a fixed background connection, the natural set of gauge-invariant observables are the open Wilson lines

$$\mathrm{Tr}_{\mathsf{R}_L} \mathcal{P} \exp\left( \int_\gamma A_L \right) \mathrm{Tr}_{\mathsf{R}_R} \mathcal{P} \exp\left( -\int_\gamma A_R \right) \tag{2.31}$$

associated to an conjugacy class $[\gamma]$ of $\Gamma$ (equivalently, a free closed loop in $M$, up to homotopy), and a pair, $\mathsf{R}_{L/R}$, of $\mathfrak{sl}(2,\mathbb{R})$ representations.

A conjugacy class $[\gamma]$ comes equipped with an inherent orientation that then defines the path-ordering appearing in (2.31). In what follows it will also be useful to work with *unoriented* free loops. To be explicit we define the following sets:

$$[\Gamma] \equiv \left\{ [\gamma] \;\Big|\; \gamma \in \Gamma \, , \; \gamma \neq 1 \right\} \, , \qquad [\Gamma]_+ \equiv [\Gamma]/\mathbb{Z}_2 \, , \tag{2.32}$$

where the $\mathbb{Z}_2$ action sends $[\gamma]$ to $[\gamma^{-1}]$. The set[10] $[\Gamma]$ is in correspondence with all non-contractible oriented free loops in $M$, while $[\Gamma]_+$ corresponds to non-contractible unoriented free loops. As described in the previous section, once we specify an ambient orientation of $M$, the holonomies of background connections can be used to assign a complex length to any $[\gamma] \in [\Gamma]$ via (2.24):

$$[\gamma] \longmapsto \mathrm{Tr}_{\mathsf{R}_L} \mathcal{P} \exp\left( \oint_\gamma A_L \right) \mathrm{Tr}_{\mathsf{R}_R} \mathcal{P} \exp\left( -\oint_\gamma A_R \right) = \mathrm{Tr}_{\mathsf{R}_L}\left( q_\gamma^{L_0} \right) \mathrm{Tr}_{\mathsf{R}_R}\left( \bar{q}_\gamma^{\bar{L}_0} \right) \, , \tag{2.33}$$

with with either $|q_\gamma| < 1$ or $|q_\gamma| > 1$ (i.e. either positive or negative $l_\gamma$, respectively) depending on if its orientation aligns or anti-aligns, respectively, with the ambient orientation of $M$. Once $[\gamma]$ has been assigned a complex length, $\hat{l}_\gamma$, via (2.33), then $[\gamma^{-1}]$ is assigned $-\hat{l}_\gamma$.

The association (2.33) induces a corresponding map on $[\Gamma]_+$ in which we associate both $\pm\hat{l}_\gamma$ to each conjugacy class as well as its inverse class. Within this association we are free to pick the representative and in what follows we will always pick the representative of $[\gamma]_+ \in [\Gamma]_+$ to map to a holonomy with $|q_\gamma| < 1$. In words, we will choose the association of $[\Gamma]_+$ to Wilson loop observables as the following: given a conjugacy class $[\gamma]$, and an ambient orientation of $M$, we compute Wilson loops à la (2.33), inputting either $[\gamma]$ or $[\gamma^{-1}]$ appropriate such that the geodesic length is positive.

## 2.3 Coupling in matter

We now consider the problem of coupling matter to the geometry. We will focus on the theory of a minimally coupled massive spin-$\mathsf{s}$ field. This is described by a symmetric $\mathsf{s}$-tensor, $\Phi_{\mu_1\mu_2\ldots\mu_\mathsf{s}}$, and a tower of associated Stückelberg fields that enforce transverse and

---

[9]If $\pi : \mathbb{H}^3 \rightarrow M$ is the natural projection, then the gauge fields on $\mathbb{H}^3$ are just the pullbacks $\pi^* A_{L/R}$.

[10]Note that $[\Gamma]$ is a set instead of a group since it is composed of conjugacy classes as opposed to group elements.

traceless conditions [21]:

$$\nabla^\nu \Phi_{\nu\mu_2\dots\mu_s} = \Phi^\nu{}_{\nu\mu_3\dots\mu_s} = 0 \ . \tag{2.34}$$

The local path integral is then given by the functional determinant

$$Z^M_{\Delta,s} = \det\left(-\nabla^2_{(s)} + \bar{m}^2_s \ell^2\right)^{-1/2} \ , \tag{2.35}$$

where $\nabla^2_{(s)}$ is the Laplace-Beltrami operator of $M$ acting on symmetric-transverse-traceless (STT) $s$-tensors, and $\bar{m}_s$ is the effective mass [22]. We have labeled the path integral by a conformal dimension, $\Delta$, related to $\bar{m}_s$ via [23]

$$\bar{m}^2_s \ell^2 = \Delta(\Delta - 2) - s \ , \tag{2.36}$$

and is the conformal dimension of a dual conformal primary through the AdS/CFT dictionary. We can relate this to $\mathfrak{sl}(2,\mathbb{R})$ representation theory in the following way.

We can realize the Laplace-Beltrami operator as the Casimir of $\mathfrak{sl}(2,\mathbb{R})$ vector fields [23],

$$c_2^{\mathfrak{sl}(2,\mathbb{R})_L} + c_2^{\mathfrak{sl}(2,\mathbb{R})_R} = \frac{1}{2}\left(\nabla^2_{(s)} + s(s+1)\right) \ , \tag{2.37}$$

and so on-shell states are states of a representation $\mathcal{R}_{\Delta,s}$ of $\mathfrak{sl}(2,\mathbb{R})_L \oplus \mathfrak{sl}(2,\mathbb{R})_R$ satisfying

$$\left(c_2^{\mathfrak{sl}(2,\mathbb{R})_L} + c_2^{\mathfrak{sl}(2,\mathbb{R})_R}\right)|\psi\rangle = \frac{1}{2}\left(\Delta(\Delta-2) + s^2\right)|\psi\rangle, \quad \forall \, |\psi\rangle \in \mathcal{R}_{\Delta,s} \ . \tag{2.38}$$

The representations satisfying this 'mass-shell condition' are pairs $(\mathsf{R}_L \otimes \mathsf{R}_R) \in \mathcal{R}_{\Delta,s}$ with

$$\begin{aligned}
\mathcal{R}^{\mathrm{HW}}_{\Delta,s} &= \{\, \mathsf{R}^{\mathrm{HW}}_{j_+} \otimes \mathsf{R}^{\mathrm{HW}}_{j_-}, \mathsf{R}^{\mathrm{HW}}_{j_-} \otimes \mathsf{R}^{\mathrm{HW}}_{j_+} \,\} \ , \\
\mathcal{R}^{\mathrm{LW}}_{\Delta,s} &= \{\, \mathsf{R}^{\mathrm{LW}}_{j_+} \otimes \mathsf{R}^{\mathrm{LW}}_{j_-}, \mathsf{R}^{\mathrm{LW}}_{j_-} \otimes \mathsf{R}^{\mathrm{LW}}_{j_+} \,\} \ ,
\end{aligned} \tag{2.39}$$

where $\mathsf{R}^{\mathrm{HW/LW}}_{j_\pm}$ are highest/lowest-weight representations[11] of $\mathfrak{sl}(2,\mathbb{R})$ with highest/lowest weight related to the mass and spin through

$$j_\pm = \frac{\Delta \pm s}{2} \ . \tag{2.40}$$

Note that for scalars, $s = 0$, $j_\pm$ collide and the set of representations is effectively halved:

$$\mathcal{R}^{\mathrm{HW/LW}}_{\Delta,0} = \{\, \mathsf{R}^{\mathrm{HW/LW}}_{\frac{\Delta}{2}} \otimes \mathsf{R}^{\mathrm{HW/LW}}_{\frac{\Delta}{2}} \,\} \ . \tag{2.41}$$

See Appendix A for details on this set of representations and how they are built.

One relevant aspect of these representations is that given a basis, $\{L_\pm, L_0\}$, of $\mathfrak{sl}(2,\mathbb{R})$ satisfying

$$[L_\pm, L_0] = \pm L_\pm \ , \qquad [L_+, L_-] = 2L_0 \ , \tag{2.42}$$

---

[11]The relevance of highest/lowest weight representations will be further discussed in Section 3.2.3.

then their characters,

$$\chi_j^{\text{HW/LW}}(\tau) \equiv \text{Tr}_{\mathsf{R}_j^{\text{HW/LW}}} \left( q^{L_0} \right) , \qquad q = e^{i2\pi\tau} , \tag{2.43}$$

are given by

$$\chi_j^{\text{HW}}(\tau) = \frac{q^{-j}}{1 - q^{-1}} = \frac{e^{-i\pi(2j-1)\tau}}{2\sinh(i\pi\tau)} , \qquad \chi_j^{\text{LW}}(\tau) = \frac{q^{j}}{1 - q} = \frac{e^{i\pi(2j-1)\tau}}{2\sinh(-i\pi\tau)} . \tag{2.44}$$

Strictly speaking, the highest weight character converges for $|q| > 1$ ($\tau$ belonging to the lower-half plane), while the lowest-weight character converges for $|q| < 1$ ($\tau$ belonging to the upper-half plane).

# 3 The Wilson spool for smooth hyperbolic quotients

In this section we more concretely introduce our main result and evidences supporting it. In [7], it was shown that the path integral of a massive spinning field on a BTZ background could be expressed as a line operator that wraps the black hole horizon arbitrarily many times. Here we extend this result to the any smooth hyperbolic quotient, showing that that the one-loop determinant appearing in (2.35) takes the form of a line operator wrapping cycles of the background topology. We will state the result first and then show that it passes a non-trivial on-shell check. We will then give several physical derivations in support of our result.

Let $M$ be a smooth, cusp-free, hyperbolic three manifold which is diffeomorphic to $\mathbb{H}_3/\Gamma$. Then

$$\log Z_{\Delta,\mathsf{s}}^M[g_{\mu\nu}] = \mathbb{W}_\Gamma[A_L, A_R] , \tag{3.1}$$

where

$$\mathbb{W}_\Gamma = \sum_{[\Gamma]_+} \sum_{\mathcal{R}_{\Delta,\mathsf{s}}^{\text{LW}}} \frac{1}{n_\gamma} \left[ \text{Tr}_{\mathsf{R}_L} \mathcal{P} \exp \left( \oint_\gamma A_L \right) \right] \left[ \text{Tr}_{\mathsf{R}_R} \mathcal{P} \exp \left( -\oint_\gamma A_R \right) \right] \tag{3.2}$$

is the Wilson spool generalized to hyperbolic quotients. Per the discussion in the previous section, the sum over $[\Gamma]_+$ corresponds to a sum over unoriented non-contractible loops in $M$ which by convention have a positive geodesic length assigned to them. This sum is weighted by that element's multiplicity. In Section 3.2.2 we will give a geometric interpretation to this factor. These Wilson loops are taken over lowest-weight representations $\mathsf{R}_L \otimes \mathsf{R}_R \in \mathcal{R}_{\Delta,\mathsf{s}}^{\text{LW}}$ appearing in (2.39). Per the previous section these representations have convergent characters when $\gamma$ has positive geodesic length.

Because every $\gamma$ can be written as $\gamma = (\gamma_0)^{n_\gamma}$ for a unique (up to conjugation) primitive $\gamma_0$ we can alternatively express $\mathbb{W}_\Gamma$ in the integral form

$$\mathbb{W}_\Gamma = \frac{i}{2} \sum_{[\Gamma_0]_+} \sum_{\mathcal{R}_{\Delta,\mathsf{s}}^{\text{LW}}} \int_{\mathcal{C}} \frac{d\alpha}{\alpha} \frac{\cos\alpha/2}{\sin\alpha/2} \left[ \text{Tr}_{\mathsf{R}_L} \mathcal{P} \exp \left( \frac{\alpha}{2\pi} \oint_{\gamma_0} A_L \right) \right] \left[ \text{Tr}_{\mathsf{R}_R} \mathcal{P} \exp \left( -\frac{\alpha}{2\pi} \oint_{\gamma_0} A_R \right) \right] , \tag{3.3}$$

where now the sum is over $[\gamma_0]_+ \in [\Gamma_0]_+$ which, in complete analogy to (2.32), corresponds to the set of unoriented primitive loops of $M$. The $\alpha$ integration contour $\mathcal{C}$, depicted in Figure 4, runs clockwise below and above the positive $\text{Re}(\alpha)$ axis, crossing just to right of the origin. Note that by construction $\Gamma$ only contains hyperbolic or loxodromic elements and so $|q_\gamma| \neq 1$. Thus given the characters (2.44), on-shell Wilson loops contribute poles sitting off the $\text{Re}(\alpha)$ axis and the contour only picks up the poles at $\alpha \in \mathbb{N}$ coming from the $\cot \alpha/2$.

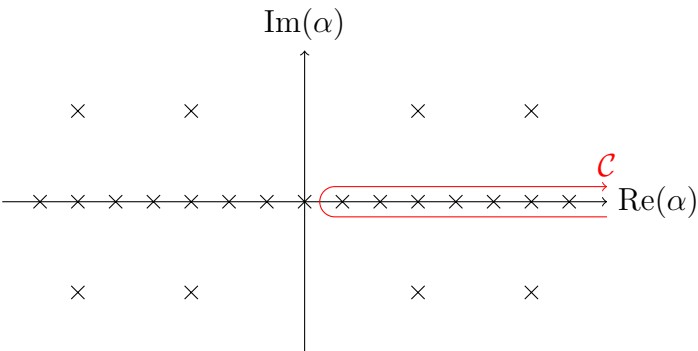

**Figure 4:** The $\alpha$ integration contour wraps the poles of $\cot \alpha/2$ lying along the real axis. Poles from the on-shell characters always appear off the real axis owing to the loxodromic or hyperbolic structure of group element, $\gamma$.

While (3.3) seems to be a somewhat inconsequential rewriting of (3.2), it serves the benefit of both uniformizing it with previous expressions of the Wilson spool appearing in [5–8] (where the integral form plays a more key role) as well as emphasizing that there is a Wilson spool for each primitive loop of $M$. In Section 4 we will discuss potential scenarios where the pole structure makes the integral form (3.3) more natural.

### 3.1 Comparison to Giombi, Maloney, and Yin

Before deriving the expression (3.2), we will demonstrate that it produces known results in the literature for the one-loop determinant of matter fields around an on-shell geometry [14].

Let $a_L, a_R$ correspond to the flat gauge connections which correspond to the on-shell hyperbolic metric on $M$ through the identification (2.5). Any flat $\text{SL}(2, \mathbb{C})$ connection on $M$ is determined, up to gauge equivalence, by its holonomies around the closed loops on $M$. The on-shell gauge field $a_L$ is precisely the flat connection such that

$$\mathcal{P} \exp \left( \oint_\gamma a_L \right) = U(p)^{-1} \gamma U(p), \tag{3.4}$$

for some periodic element, $U(p) \in \text{SL}(2, \mathbb{C})$, and by abuse of notation, we use $\gamma$ to refer both to a loop $\gamma \in \pi_1(M, p)$ with respect to a base point $p$, as well as an element $\gamma \in \Gamma$. Since $a_R = -\overline{a_L}$, we also have

$$\mathcal{P} \exp \left( -\oint_\gamma a_R \right) = U(p)^\dagger \gamma^\dagger (U(p)^\dagger)^{-1}. \tag{3.5}$$

Now we use the assumption that $\gamma$ is either hyperbolic or loxodromic, which is necessary to ensure that the quotient $\mathbb{H}^3/\Gamma$ is smooth (see Appendix B). This means that it can be diagonalized into the form

$$\gamma = \begin{pmatrix} q_\gamma^{1/2} & 0 \\ 0 & q_\gamma^{-1/2} \end{pmatrix} = e^{2\pi i \tau_\gamma L_0}, \tag{3.6}$$

with $q_\gamma = e^{2\pi i \tau_\gamma}$. Together with (3.4), (3.5) this recovers the on-shell statement of the length/holonomy relationship (2.24).

Knowing these holonomies is sufficient to compute the on-shell value of $\mathbb{W}_\Gamma$. Indeed, the Wilson loop along $\gamma$ now just computes the character of $\gamma$ in the appropriate representation:

$$\left[ \mathrm{Tr}_{\mathsf{R}_L} \mathcal{P} \exp \left( \oint_\gamma a_L \right) \right] \left[ \mathrm{Tr}_{\mathsf{R}_R} \mathcal{P} \exp \left( -\oint_\gamma a_R \right) \right] = \chi_{\mathsf{R}_L}(\tau_\gamma) \overline{\chi_{\mathsf{R}_R}(\tau_\gamma)}, \tag{3.7}$$

for some $\tau_\gamma$ which, by choice of representative of $[\gamma]_+$ in $[\Gamma]_+$, we take to live in the upper-half plane. The character of the representative of $[\gamma]_+$ in the representations appearing in $\mathcal{R}_{\Delta,s}^{\mathrm{LW}}$ are now readily computed from (2.44):

$$\sum_{\mathcal{R}_{L/R}^{\mathrm{LW}}} \left[ \mathrm{Tr}_{\mathsf{R}_L} \mathcal{P} \exp \left( \oint_\gamma a_L \right) \right] \left[ \mathrm{Tr}_{\mathsf{R}_R} \mathcal{P} \exp \left( -\oint_\gamma a_R \right) \right] = \frac{q_\gamma^{j+} \bar{q}_\gamma^{j-} + q_\gamma^{j-} \bar{q}_\gamma^{j+}}{|1 - q_\gamma|^2}. \tag{3.8}$$

We can now compute the on-shell value of the Wilson spool (1.3) by summing over all representatives, $[\gamma]_+$ of $[\Gamma]_+$. As alluded to before we can decompose this into a sum over representatives, $[\gamma_0]_+$, of primitive generators and integers $n_\gamma$ such that $[\gamma]_+ = \left[\gamma_0^{n_\gamma}\right]_+$. Explicitly,

$$\sum_{[\Gamma]_+} \sum_{\mathcal{R}_{\Delta,s}^{\mathrm{LW}}} \frac{1}{n_\gamma} \left[ \mathrm{Tr}_{\mathsf{R}_L} \mathcal{P} \exp \left( \oint_\gamma a_L \right) \right] \left[ \mathrm{Tr}_{\mathsf{R}_R} \mathcal{P} \exp \left( -\oint_\gamma a_R \right) \right]$$

$$= \sum_{[\Gamma_0]_+} \sum_{\mathcal{R}_{\Delta,s}^{\mathrm{LW}}} \sum_{n_\gamma=1}^\infty \frac{1}{n_\gamma} \left[ \mathrm{Tr}_{\mathsf{R}_L} \mathcal{P} \exp \left( \oint_{\gamma_0^n} a_L \right) \right] \left[ \mathrm{Tr}_{\mathsf{R}_R} \mathcal{P} \exp \left( -\oint_{\gamma_0^n} a_R \right) \right] \tag{3.9}$$

$$= \sum_{[\gamma_0]_+ \neq 1} \sum_{n=1}^\infty \frac{1}{n} \frac{q_{\gamma_0}^{nj+} \bar{q}_{\gamma_0}^{nj-} + q_{\gamma_0}^{nj-} \bar{q}_{\gamma_0}^{nj+}}{|1 - q_{\gamma_0}^n|^2}.$$

Expanding the denominator as a geometric series $|1-q_{\gamma_0}^n|^2 = \sum_{\ell,\bar{\ell}=0}^\infty q_{\gamma_0}^{n\ell} \bar{q}_{\gamma_0}^{n\bar{\ell}}$, we can perform the sum over $n$ and we are left with the final result:

$$\exp\left( \mathbb{W}_\Gamma \big|_{\mathrm{on\text{-}shell}} \right) = \prod_{[\gamma_0]_+ \neq 1} \prod_\pm \prod_{\ell,\bar{\ell}=0}^\infty \left( 1 - q_{\gamma_0}^{\ell + \frac{\Delta \pm s}{2}} \bar{q}_{\gamma_0}^{\bar{\ell} + \frac{\Delta \mp s}{2}} \right)^{-1}. \tag{3.10}$$

For the values $s = 0, 1$, this reproduces the known one-loop determinants of spinning massive matter[12] on $\mathbb{H}^3/\Gamma$ computed by Giombi, Maloney, and Yin (GMY) [14]. To our knowledge, up to now the corresponding one-loop determinants for massive $s \geq 2$ fields on $\mathbb{H}^3/\Gamma$ have not been constructed; the on-shell spool (3.10) fills this gap.

## 3.2 Derivations of the main result

Having demonstrated the validity of the spool for massive scalars and vectors by comparison to [14], we now present a series of more general derivations. We firstly perform a derivation of the one-loop determinant for minimally coupled fields of any spin $s$ on a cusp-free hyperbolic manifold via the Selberg trace formula. While this derivation ostensibly only holds for 'on-shell' metrics, it will nonetheless ultimately result in the gauge invariant formulation of Wilson loops in the appropriate representations in agreement with (1.3).

Our subsequent two derivations will provide further evidence for the interpretation of (3.1) as an off-shell statement. Firstly we will express the one-loop determinant of a scalar field in the worldline path integral formalism ultimately relating elements of this calculation to the elements of the Wilson spool; we will perform this path integral on a 'perturbatively off-shell' manifold, i.e. one that that is topologically $\mathbb{H}^3/\Gamma$ but with a potentially off-shell metric. We then demonstrate, in a similar manner to [14], that this can be reduced into a sum over conjugacy classes of worldline path integrals of the quotient space. One consequence of this analysis is the emphasis that decomposition of loops into conjugacy classes and the multiplicity, $n_\gamma$, is a topological statement and is sensible off-shell.

Finally we revisit the quasinormal mode method [24] for constructing the one-loop determinant on the torus [6, 7] and extend that analysis to smooth hyperbolic quotients. While, even for simple geometries, explicit quasinormal modes are only known for on-shell metrics, Appendix D of [5] illustrates how the scalar quasinormal spectra can be organized into representation theory of the local isometries even for perturbatively off-shell metrics. The assumption that this holds for the massive spinning spectra then gives our construction an off-shell interpretation.

### 3.2.1 The Selberg Trace

The Selberg Trace formula[13] provides us with a relationship between two spectra on hyperbolic manifolds:

- The spectrum of the Laplacian, which generically contains both discrete and continuous components. Knowledge of this spectrum is sufficient to reconstruct the one-loop determinant on the quotient space.

---

[12]While strictly speaking, from a physical perspective $\mathbb{W}_\Gamma$ should only apply to massive fields, we also note that (3.10) gives the one-loop determinant for a massless, STT spin-2 particle when setting $\Delta = s = 2$.

[13]For an introduction to the Selberg trace formula for the scalar Laplacian see [25].

- The complex length spectrum of closed geodesics. In Section 2.1.1 we showed how such information can be interpreted in the language of Chern-Simons theory.

On the 'spectral' side of the trace formula we label the eigenvalues of the spin-$s$ traceless, transverse, divergence free Laplacian by $\lambda_m^{(s)}$ which we write as $\lambda_m^{(s)} = (t_m^{(s)})^2 + s + 1$. We will formulate this for compact quotients composed entirely of hyperbolic and loxodromic elements so that this spectrum is rendered discrete. On the 'geometric' side, we have conjugacy classes $[\gamma]$ of elements $\gamma \in \Gamma$ which, as discussed several times above, correspond to a homotopy classes of free loops with associated complex lengths, $\hat{l}_\gamma = l_\gamma + i\theta_\gamma$.

Then given an even test function, $H : \mathbb{R} \to \mathbb{R}$, the Selberg trace formula relates these two spectra as [26][14][15]

$$(1 + \delta_{s,0}) \sum_m \widehat{H}(t_m^{(s)}) = \frac{\text{vol}(\mathbb{H}^3/\Gamma)}{\pi} \left( s^2 H(0) - H''(0) \right) + 2 \sum_{[\Gamma]_+} \frac{l_\gamma}{n_\gamma} \frac{\cos(s\,\theta_\gamma) H(l_\gamma)}{\cosh l_\gamma - \cos\theta_\gamma} \ , \quad (3.11)$$

where $\widehat{H}$ is the Fourier transform of $H$,

$$\widehat{H}(t) = \int_{-\infty}^{\infty} \mathrm{d}x\, H(x) e^{-ixt} \ . \tag{3.12}$$

The clever choice of test function

$$H(x) = \frac{1}{\sqrt{4\pi\beta}} e^{-\frac{x^2}{4\beta}} \ , \qquad \widehat{H}(t) = e^{-\beta t^2} \ , \tag{3.13}$$

then reduces the Selberg trace formula to

$$\sum_m e^{-\beta \lambda_m^{(s)}} = \frac{2 - \delta_{s,0}}{2} e^{-\beta(1+s)} \left[ 2 \frac{\text{vol}(\mathbb{H}^3/\Gamma)}{(4\pi\beta)^{\frac{3}{2}}} (1 + 2\beta s^2) \right.$$
$$\left. + \sum_{[\Gamma]_+} \frac{l_\gamma}{n_\gamma} \frac{\exp\left(-\frac{l_\gamma^2}{4\beta}\right)}{\sqrt{4\pi\beta}} \frac{\cos(s\theta_\gamma)}{\sinh\frac{\hat{l}_\gamma}{2} \sinh\frac{\hat{l}_\gamma^*}{2}} \right] \ . \tag{3.14}$$

This sum contains information about the full spectrum of the Laplacian and can be used to reconstruct the one-loop determinant

$$\log\det\left(-\nabla^2_{(s)} + \bar{m}_s^2 \ell^2\right) = -\int_0^\infty \frac{\mathrm{d}\beta}{\beta} e^{-\beta \bar{m}_s^2 \ell^2} \sum_m e^{-\beta \lambda_m^{(s)}} \ , \tag{3.15}$$

Noting that $l_\gamma$ is strictly positive in the case of a cusp-free hyperbolic manifold this integral can be exactly computed:

---

[14]Within [26] both the scalar and spin-1 trace formulae include a contribution due to the trivial representation of $\text{PSL}(2,\mathbb{C})$ identified with constant functions on a compact manifold. We have absorbed all such considerations into the sum over eigenvalues.

[15]This formula is usually stated as a sum over non-trivial conjugacy classes $[\gamma] \neq 1$ implicitly assigning a positive geodesic length to each $[\gamma]$. In our notation this is equivalent to twice the summation over $[\Gamma]_+$.

$$\log \det \left(-\nabla^2_{(\mathsf{s})} + \bar{m}^2_{\mathsf{s}}\ell^2\right) = -(2 - \delta_{\mathsf{s},0})\mathrm{vol}(\mathbb{H}^3/\Gamma) \int_0^\infty \frac{\mathrm{d}\beta}{\beta} \frac{e^{-\nu^2\beta}}{(4\pi\beta)^{\frac{3}{2}}} \left[1 + 2\beta\mathsf{s}^2\right]$$

$$- \frac{2 - \delta_{\mathsf{s},0}}{4} \sum_{[\Gamma]_+} \frac{1}{n_\gamma} \frac{\exp\left(-\nu l_\gamma\right)}{\sinh \frac{\hat{l}_\gamma}{2} \sinh \frac{\hat{l}^*_\gamma}{2}} \left(e^{i\mathsf{s}\theta_\gamma} + e^{-i\mathsf{s}\theta_\gamma}\right) \ , \quad (3.16)$$

with $\nu^2 = \bar{m}^2_{\mathsf{s}}\ell^2 + \mathsf{s} + 1$.

The first line of (3.16) contains a UV divergent contribution to the one-loop determinant coming from the $\beta \sim 0$ behavior of the integral. Schematically we can view this divergence as the contribution of identity class of $\Gamma$ (i.e. loops contractible to a point) and thus scaling like the volume of $M$. In order to make sense of this term we must regulate the $\beta$ integral near 0 and prescribe a renormalization condition for removing the divergence. Regardless, this contribution is non-universal and dependent on the details of renormalization. In what follows we will follow the simple prescription of subtracting it off to define

$$\log Z_{\Delta,\mathsf{s}} = -\frac{1}{2} \log \det \left(-\nabla^2_{(\mathsf{s})} + \bar{m}^2_{\mathsf{s}}\ell^2\right)\Big|_{\mathrm{ren.}}$$

$$\equiv \frac{2 - \delta_{\mathsf{s},0}}{8} \sum_{[\Gamma]_+} \frac{1}{n_\gamma} \frac{\left(e^{-\nu l_\gamma + i\mathsf{s}\theta_\gamma} + e^{-\nu l_\gamma - i\mathsf{s}\theta_\gamma}\right)}{\sinh \frac{\hat{l}_\gamma}{2} \sinh \frac{\hat{l}^*_\gamma}{2}} \ . \quad (3.17)$$

Our final step in casting this as a Chern-Simons observable is to recall the relation between complex lengths and holonomies as outlined in Section 2.1.1 and recognize the fraction as a product of lowest-weight characters from Section 2.3. All-in-all we find

$$\log Z_{\Delta,\mathsf{s}} = \sum_{[\Gamma]_+} \sum_{\mathcal{R}^{\mathrm{LW}}_{\Delta,\mathsf{s}}} \frac{1}{n_\gamma} \left[\mathrm{Tr}_{\mathsf{R}_L} \mathcal{P} \exp\left(\oint_\gamma A_L\right)\right] \left[\mathrm{Tr}_{\mathsf{R}_R} \mathcal{P} \exp\left(-\oint_\gamma A_R\right)\right] \ . \quad (3.18)$$

Let us now make a couple of comments.

Firstly this result is consistent with the derivation in GMY [14] in the cases of scalar and massive vector fields. In fact the GMY result holds more generally for non-compact quotients without cusps, such as thermal AdS. While this is only demonstrated explicitly for massive fields up to $\mathsf{s} = 1$, our casting of the Selberg trace as (3.18) suggests it extends to all spinning fields, even for non-compact quotients.

Secondly, while the above derivation applies strictly to the on-shell complete hyperbolic metric, the casting of it as Chern-Simons variables (and more specifically as their gauge-invariant holonomies) is natural to take off-shell and to capture metric fluctuations within the gravitational path-integral. In the following section we will give credence to this while also emphasizing the topological origin of the multiplicity factor, $n_\gamma$.

### 3.2.2 The worldline path integral

In this section we make more explicit the physical intuition for our result by demonstrating how the scalar one-loop determinant can be recovered from a path integral calculation;

in doing so we will also make explicit that the decomposition into conjugacy classes and the factor $n_\gamma$ are topological features. To be clear on scope, we will consider the worldline path integral of a scalar field over free loops in $\mathbb{H}^3/\Gamma$ treated simply as a smooth manifold, with a potentially off-shell metric $g$. This can be divided into disjoint sectors of different free homotopy classes, which are in one-to-one correspondence with the conjugacy classes of $\Gamma$. We will then show that by lifting our calculation from $\mathbb{H}^3/\Gamma$ to $\mathbb{H}^3$ the path integral can be reorganized into a sum over conjugacy classes, each being simply an integral over a torus topology. In practice, to perform a lift to $\mathbb{H}^3$ it is necessary to 'break' the loop at a point, and this temporarily reintroduces a sum over all possible group elements within each conjugacy class.

Finally we shall show how taking the metric on-shell to the hyperbolic metric on $\mathbb{H}^3/\Gamma$ reproduces exactly the previous results and allows a matching of terms to the on-shell Wilson spool. This follows due to the two-loop exactness of this calculation within perturbation theory.

Following [27, 28] we can write the scalar one-loop determinant as a worldline path integral

$$\log\det\left(-\nabla^2 + m^2\ell^2\right)^{-1/2} = \frac{1}{2}\int_0^\infty \frac{\mathrm{d}\beta}{\beta} e^{-\beta m^2\ell^2}\, \mathcal{K}(\beta)\,, \tag{3.19}$$

with

$$\mathcal{K}(\beta) \equiv \int\limits_{x(0)=x(\beta)} \mathcal{D}x\, \exp\left(-\frac{1}{4}\int_0^\beta \mathrm{d}s\, g_{ab}\dot{x}^a\dot{x}^b\right)\,. \tag{3.20}$$

A couple of notes about this worldline path integral. Firstly, it is defined intrinsically on the quotient manifold $\mathbb{H}^3/\Gamma$, and the measure $\mathcal{D}x$ is the sum over closed loops *with* initial point $x(0)$.[16] Secondly, so far within this path integral we are considering the smooth manifold $\mathbb{H}^3/\Gamma$ without necessarily choosing the hyperbolic metric upon it; this potentially could be a perturbatively off-shell geometry within the gravitational path integral.

We now lift our calculation onto the covering space $\mathbb{H}^3 \xrightarrow{\pi} \mathbb{H}^3/\Gamma$. Again we think of $\mathbb{H}^3$ only as a smooth manifold, equipped with the potentially off-shell metric $\tilde{g} = \pi^*(g)$ periodic under the action of $\Gamma$ on $\mathbb{H}^3$. We additionally lift each path $x$ in $\mathbb{H}^3/\Gamma$ uniquely to an *open* curve $\tilde{x}$ in $\mathbb{H}^3$ with $\tilde{x}(0) \in \mathcal{F}$, a fundamental domain of $\Gamma$ in $\mathbb{H}^3$. Importantly, closed loops in the same free homotopy class of $\mathbb{H}^3/\Gamma$ may lift to open curves in $\mathbb{H}^3$ which are not homotopic. Open curves may end at $\tilde{x}(\beta) = \tilde{\gamma}\tilde{x}(0)$ for any $\tilde{\gamma} \in [\gamma]$. In effect we have divided each free conjugacy class into the based conjugacy classes comprising it. We write for the non-identity sectors of the path integral:

$$\mathcal{K}(\beta) = \sum_{[\gamma]\neq 1} \sum_{\tilde{\gamma}\in[\gamma]} \int\limits_{\substack{\tilde{x}(0)\in\mathcal{F} \\ \tilde{x}(\beta)=\tilde{\gamma}\tilde{x}(0)}} \mathcal{D}x\, \exp\left(-\frac{1}{4}\int_0^\beta \mathrm{d}s\, \tilde{g}_{ab}\dot{\tilde{x}}^a\dot{\tilde{x}}^b\right)\,. \tag{3.21}$$

---

[16]We can think of the factor of $\frac{1}{\beta}$ in this integral as accounting for the gauge redundancy of fixing the initial point on a closed loop.

We can now utilize a trick in [14] and note that summing over all conjugates of $\gamma$ on fundamental domain $\mathcal{F}$ is equivalent to considering only the representative $\gamma$ but integrating over the fundamental domain $\mathcal{F}_{\mathcal{C}(\gamma)}$, the fundamental domain for the centralizer of $\gamma$ in $\Gamma$. That is, subgroups of $\Gamma$ preserving the same sets of fixed points can be treated totally independently.

Finally we can observe that when $\mathbb{H}^3/\Gamma$ is a cusp-free smooth manifold, $\Gamma$ is torsion-free and $\mathcal{C}(\gamma)$ is simply an infinite cyclic group (see Appendix B). As such the fundamental domain for the subgroup generated purely by $\gamma$, denoted $\mathcal{F}_{\langle\gamma\rangle}$, is simply $n_\gamma$ copies of the fundamental domain $\mathcal{F}_{\mathcal{C}(\gamma)}$:

$$
\mathcal{K}(\beta) = \sum_{[\gamma]\neq 1} \mathcal{K}_{[\gamma]} \equiv \sum_{[\gamma]\neq 1} \frac{1}{n_\gamma} \int_{\substack{\tilde{x}(0)\in\mathcal{F}_{\langle\gamma\rangle} \\ \tilde{x}(\beta)=\gamma\tilde{x}(0)}} \mathcal{D}\tilde{x} \, \exp\left(-\frac{1}{4}\int_0^\beta \mathrm{d}s\, \tilde{g}_{ab}\dot{\tilde{x}}^a\dot{\tilde{x}}^b\right) . \tag{3.22}
$$

This path integral is equivalent a saddle-point sum over path-integrals, $\mathcal{K}_{[\gamma]}$, each of which is the path integral over a torus with generator $\gamma$ and an off-shell metric $\tilde{g}$. We see that any off-shell one-loop determinant on $\mathbb{H}^3/\Gamma$ can be reduced to a sum over conjugacy classes and an off-shell one-loop determinant on the torus. In particular this sum and the factor $n_\gamma$ need not be varied for off-shell geometries within a gravitational path integral; they are properties of the topology of the quotient.

For completeness let us also now show that path integral methods can reproduce the correct functional form when the metric $g$ is taken to be the on-shell hyperbolic metric. Any hyperbolic or loxodromic element can be put into the form (2.28) with $q_\gamma = \exp \hat{l}_\gamma$. Considering the $\mathcal{K}_{[\gamma]}$ contribution, without loss of generality we can thus take a choice of coordinates on $\mathbb{H}^3$ such that

$$
\tilde{g}_{ab}\mathrm{d}\tilde{x}^a\mathrm{d}\tilde{x}^b = (1+r^2)\,\mathrm{d}t^2 + \frac{\mathrm{d}r^2}{1+r^2} + r^2\,\mathrm{d}\phi^2 , \tag{3.23}
$$

with $t \in \mathbb{R}$, $r \in [0,\infty)$, $\phi \in [0,2\pi)$. The transformation generated by $\gamma$ maps

$$
\begin{aligned}
t &\to t + l_\gamma , \\
\phi &\to \phi + \theta_\gamma ,
\end{aligned} \tag{3.24}
$$

and so a suitable fundamental domain $\mathcal{F}_{\langle\gamma\rangle}$ is $t \in [0,l_\gamma)$, $r \in [0,\infty)$, $\phi \in [0,2\pi)$. For readability in this section we shall henceforth drop the explicit $\gamma$ notation for complex length parameters, restoring it at the end.

The minimal length geodesic for the transformation generated by $\gamma$ runs along $r=0$, and as an extremum of the worldline path integral we would like to perform a saddle-point approximation around it. It will be convenient to work in a Cartesian version of this metric as in [27] with coordinates $(t,\vec{q})$ and $\vec{q} = (r\cos\phi, r\sin\phi)$:

$$
\tilde{g}_{ab}\mathrm{d}\tilde{x}^a\mathrm{d}\tilde{x}^b = (1+q^2)\,\mathrm{d}t^2 + \mathrm{d}q^2 - \frac{(q\cdot\mathrm{d}q)^2}{1+q^2} . \tag{3.25}
$$

The worldline action, $I[\tilde{x}] = \frac{1}{4}\int_0^\beta ds\, \tilde{g}_{ab}\dot{\tilde{x}}^a\dot{\tilde{x}}^b$, expanded around the geodesic saddle point $t(s) = l(\frac{s}{\beta} + u(s))$ is

$$I[\tilde{x}] = \frac{l^2}{4\beta} + \frac{1}{4}\int_0^\beta ds\left[l^2\dot{u}^2 + \dot{q}^2 + \frac{l^2}{\beta^2}q^2 + 2\frac{l^2}{\beta}\dot{u}q^2 + l^2\dot{u}^2q^2 - \frac{(q\cdot\dot{q})^2}{1+q^2}\right] . \tag{3.26}$$

There is additionally a quantum counterterm which appears due to the Weyl-ordering of the Hamiltonian. Such counterterms are well understood and following the time-slicing prescription[17] of [27, 28] we add

$$I_{\text{c.t.}}[\tilde{x}] = -\frac{\ell^2}{4}\int_0^\beta ds(\tilde{R} + \tilde{g}^{\mu\nu}\tilde{\Gamma}^\rho_{\mu\sigma}\tilde{\Gamma}^\sigma_{\nu\rho}) = \frac{3}{2}\beta . \tag{3.27}$$

The worldline path integral $\mathcal{K}_{[\gamma]}$ can be organized in a loop expansion as

$$\mathcal{K}_\gamma = \int \mathcal{D}u\mathcal{D}q\, e^{-I-I_{\text{c.t.}}} \equiv e^{-W} , \qquad W = \frac{l^2}{4\beta} + w_u^{(1)} + w_q^{(1)} + w^{(2)} + \cdots \tag{3.28}$$

Focussing on quadratic terms of $I$ first, the $u$ field is a free massless particle and (after extracting the zero mode) has a propagator

$$G_u(s_1, s_2) = \frac{(\beta - s_1)s_2}{l^2\beta}\Theta(s_1 - s_2) + s_1 \leftrightarrow s_2 , \tag{3.29}$$

with a one-loop contribution

$$w_u^{(1)}(\beta) = \frac{1}{2}\log\left(\frac{4\pi\beta}{l^2}\right) . \tag{3.30}$$

The field $q \in \mathbb{R}^2$ is little more complicated. Due to the torsion the boundary conditions on $q$ imposed by (3.24) are twisted by a rotation $R_\theta q(0) = q(\beta), R_\theta q'(0) = q'(\beta)$, where

$$R_\theta = \begin{pmatrix} \cos\theta & \sin\theta \\ -\sin\theta & \cos\theta \end{pmatrix} . \tag{3.31}$$

The Green's function equation

$$\frac{1}{2}\left(-\partial_{s_1}^2 + \frac{l^2}{\beta^2}\right)G_q(s_1, s_2) = \delta(s_1 - s_2)\mathbf{1} , \tag{3.32}$$

along with boundary conditions $R_\theta G_q(0, s_2) = G_q(\beta, s_2), R_\theta\partial_{s_1}G_q(0, s_2) = \partial_{s_1}G_q(\beta, s_2)$, are solved by

$$G_q(s_1, s_2) = \frac{\beta\,\Theta[s_2 - s_1]}{l\sinh\frac{\hat{l}}{2}\sinh\frac{\hat{l}^*}{2}}\left[\sinh\left(\frac{l}{\beta}(s_2 - s_1)\right)R_\theta^{-1} + \sinh\left(\frac{l}{\beta}(s_1 - s_2 + \beta)\right)\mathbf{1}\right]$$

---

[17]Additionally we note that within this prescription the products of distributions are defined by treating $\delta(x)$ as a Kronecker delta and by taking $\Theta(0) = \frac{1}{2}$.

$$+ \begin{pmatrix} s_1 \leftrightarrow s_2 \\ \theta \rightarrow -\theta \end{pmatrix} . \quad (3.33)$$

In order to calculate the one-loop factor for the field $q$ we can take advantage of the change of variables $q = R_{\frac{\theta s}{\beta}} Q$ which imposes periodic boundary conditions on $Q$ and thus renders the path integral over the kinetic terms in $Q$ equivalent to the thermal partition function of a quantum system with Lagrangian[18]

$$L_Q = \frac{1}{4} \dot{Q}^2 + \frac{i\theta}{2\beta} Q^T \begin{pmatrix} 0 & 1 \\ -1 & 0 \end{pmatrix} \dot{Q} - \frac{1}{4} \frac{l^2 + \theta^2}{\beta^2} Q^2 . \quad (3.34)$$

The corresponding Hamiltonian is

$$H_Q = P^2 - \frac{i\theta}{\beta} Q^T \begin{pmatrix} 0 & 1 \\ -1 & 0 \end{pmatrix} P + \frac{1}{4} \frac{l^2}{\beta^2} Q^2 , \quad (3.35)$$

which after a canonical transformation

$$\begin{pmatrix} Q \\ P \end{pmatrix} = \begin{pmatrix} 0 & 0 & -\frac{\beta}{l} & \frac{\beta}{l} \\ 1 & 1 & 0 & 0 \\ \frac{l}{2\beta} & -\frac{l}{2\beta} & 0 & 0 \\ 0 & 0 & \frac{1}{2} & \frac{1}{2} \end{pmatrix} \begin{pmatrix} \tilde{Q} \\ \tilde{P} \end{pmatrix} \quad (3.36)$$

becomes the decoupled sum of harmonic oscillator Hamiltonians of mass $l^{-1}$ and frequency $\beta^{-1}$:

$$H_Q = (l + i\theta) \tilde{H}_1 + (l - i\theta) \tilde{H}_2 , \qquad \tilde{H}_i = \frac{1}{2l} \tilde{P}_i^2 + \frac{l}{2\beta^2} \tilde{Q}_i^2 . \quad (3.37)$$

The thermal partition function of this system is then just

$$w_q^{(2)} \equiv -\log \text{Tr}\, e^{-\beta H_Q} = -\log \sum_{n_1, n_2 = 0}^{\infty} \exp\left( -\hat{l}\left(n_1 + \frac{1}{2}\right) - \hat{l}^*\left(n_2 + \frac{1}{2}\right) \right)$$

$$= \log\left( 4 \sinh\frac{\hat{l}}{2} \sinh\frac{\hat{l}^*}{2} \right) . \quad (3.38)$$

Finally we need to calculate any higher loop corrections due to the interaction terms in the action. Remarkably (due to the symmetry of the AdS$_3$ background - see [27] for some further comments) the loop expansion (3.28) is *two-loop exact*, with contributions

$$= 0 \quad (3.39)$$

---

[18]Strictly to have a real Lagrangian $\theta$ must be imaginary. We thus analytically continue our solution to real $\theta$ at the end of the calculation.

$$\ominus \qquad = \frac{\beta \sinh l - \beta l \cosh l}{l \cos \theta - l \cosh l} \qquad (3.40)$$

$$\bigcirc\!\!\!\!\bigcirc \qquad = -\frac{\beta^2 \left( \frac{1}{\beta} - \delta(0) \right) \sinh l}{l \cos \theta - l \cosh l} \qquad (3.41)$$

$$\bigcirc\!\bigcirc \qquad = \frac{\beta(\cos \theta + \cosh l)}{2(\cos \theta - \cosh l)} - \frac{\beta^2 \delta(0) \sinh l}{l \cos \theta - l \cosh l} \qquad (3.42)$$

where the solid line indicates $G_q$ and a dashed line indicates $G_u$. These diagrams sum simply to $\frac{\beta}{2}$. Combining everything together the exact effective action for the $[\gamma]$ contribution to the worldline path integral is

$$W = \frac{l^2}{4\beta} + \frac{1}{2} \log \left( \frac{4\pi\beta}{l^2} \right) + \log \left( 4 \sinh \frac{\hat{l}}{2} \sinh \frac{\hat{l}^*}{2} \right) - \frac{\beta}{2} + \frac{3\beta}{2} \ . \qquad (3.43)$$

Noting that the above assigns a positive geodesic length to each geodesic, we sum up the non-identity sector of the path integral to find

$$\log \det \left( -\nabla^2 + m^2 \ell^2 \right)^{-1/2} = \frac{1}{4} \int_0^\infty \frac{d\beta}{\beta} \sum_{[\Gamma]_+} \frac{l_\gamma}{n_\gamma} \frac{e^{-\frac{l_\gamma^2}{4\beta}}}{\sqrt{4\pi\beta}} \frac{e^{-\beta\nu^2}}{\sinh \frac{\hat{l}}{2} \sinh \frac{\hat{l}^*}{2}}$$

$$= \frac{1}{4} \sum_{[\Gamma]_+} \frac{1}{n_\gamma} \frac{e^{-\nu l_\gamma}}{\sinh \frac{\hat{l}}{2} \sinh \frac{\hat{l}^*}{2}} \ , \qquad (3.44)$$

where we recall $\nu^2 = 1 + m^2 \ell^2$ (with $\mathsf{s} = 0$). This worldline path integral precisely reproduces the renormalized one-loop determinant derived from the Selberg trace formula, (3.17), and the on-shell Wilson spool. Much like in the Selberg trace calculation, we have ignored the contribution of contractible loops to the worldline path integral. This identity contribution is subtle to recover from the approach: since $\mathcal{C}(1) = \Gamma$ we cannot use the described decomposition of the fundamental domains to simplify the fundamental domain of this sector of the worldline path integral. More importantly, this identity component is a UV divergent contribution to the one-loop determinant and so we can view the Wilson spool as computing a renormalized one-loop determinant.

The above exercise now allows us to draw some analogies between features of the Wilson spool and the worldline path integral. The Wilson spool neatly sums up the contributions of *all paths* living in the same conjugacy class which is keeping with its interpretation as a topological operator within the Chern-Simons path integral. Within each conjugacy class, the exponential damping appearing in its on-shell value can be identified with the (Laplace transform) of the corresponding saddle point contribution; within the $\mathfrak{sl}(2, \mathbb{R})$ characters this is the contribution of the lowest-weight state of $\mathsf{R}_j^{\mathrm{LW}} \otimes \mathsf{R}_j^{\mathrm{LW}}$. The $|\sinh \frac{\hat{l}}{2}|^{-2}$ denominator comes entirely from one-loop effects in the worldline

picture; for the spool this is equivalently the resummation of all the descendant states within the lowest-weight representation. Lastly the finite shift, $m^2\ell^2 \to \nu^2 = 1 + m^2\ell^2$, of the saddle point is entirety of the two-loop effect and is $\gamma$ independent; this is crucial as the corresponding shift on the spool side is representation theoretic – an ordering effect of the quadratic Casimir – as opposed to geometric.

### 3.2.3 The quasinormal mode method

In this final section we cast our Wilson spool construction in the language of quasinormal modes, which was the original setting in which the spool was derived in [5, 6]. More pertinent for the present paper, [5–7] constructed the spool for thermal AdS$_3$ through considering quasinormal modes.[19] The quasinormal mode method applied to spools follows the broad philosophy laid out in the seminal paper Denef, Hartnoll, and Sachdev (DHS) [24]; this has been thoroughly detailed in [5–7] and so we only coarsely summarize the necessary points here. In brief, this construction consists of three observations:

- The poles of $Z^2_{\Delta,\mathsf{s}}$ in the complex $\Delta$ plane, via (2.35), must align with the spin-$\mathsf{s}$ eigenvalues of the Laplacian under analytic continuation of the mass parameter.

- The corresponding eigenfunctions can be represented as weights of highest/lowest weight $\mathfrak{sl}(2,\mathbb{R})$ representations satisfying the Casimir equation, (2.38). In the BTZ background, for example, these weights correspond to the ingoing and outgoing Lorentzian quasinormal modes.

- The set of contributing eigenfunctions are further restricted by the imposition of boundary conditions, single-valuedness, and regularity.

In [7] the above properties of these eigenfunctions are codified in a group theoretic manner as a set of conditions to be satisfied by representations and their weights.

**Condition 0:** The Casimirs of the representations solve the spin-$\mathsf{s}$ mass shell condition (2.38). In locally AdS$_3$ spacetimes these are highest and lowest weight representations, $\mathsf{R}_L \otimes \mathsf{R}_R \in \mathcal{R}^{\mathrm{LW/HW}}_{\Delta,\mathsf{s}}$ with lowest/highest weight given by (2.40). Further satisfying the boundary conditions of normalizable fall-off specifies a particular root of (2.36) as is usual in the AdS/CFT dictionary.

**Condition I:** Eigenfunctions are single-valued under parallel transport around a non-trivial cycle, $\gamma$, of the geometry. It demands that a given weight in $\mathsf{R}_L \otimes \mathsf{R}_R$ has eigenvalue 1 under

$$\mathsf{R}_L \left[ \mathcal{P} \exp \left( \oint_\gamma A_L \right) \right] \mathsf{R}_R \left[ \mathcal{P} \exp \left( - \oint_\gamma A_L \right) \right] . \tag{3.45}$$

---

[19]Strictly speaking, the calculation of [6] was performed in the BTZ black hole background, but this is related to thermal AdS$_3$ by a modular S-transformation on the parameter $\tau$.

**Condition II:** Eigenfunctions must be regular everywhere on the geometry. In group theoretic terms we demand that our Lie algebra representations lift to representations of the full Lie Group. In thermal AdS$_3$ this imposes no additional restrictions on the reps $\mathcal{R}_{\Delta,\mathsf{s}}^{\mathrm{LW/HW}}$.

These conditions were shown to reproduce the one-loop determinant of thermal AdS in [6, 7]. Here we will take an alternative, slightly more schematic route to that result. Denoting the only non-trivial cycle of thermal AdS$_3$ as $\gamma_0$, we can write a formal object giving a pole for each weight satisfying **Conditions 0, I, & II** as[20]

$$Z_{\Delta,\mathsf{s}}^{\mathrm{TAdS}_3} = \prod_{\mathcal{R}_{\Delta,\mathsf{s}}} \mathrm{Det}_{\mathsf{R}_L \otimes \mathsf{R}_R} \left( 1 - \mathcal{P} \exp \left( \oint_{\gamma_0} A_L \right) \mathcal{P} \exp \left( - \oint_{\gamma_0} A_R \right) \right)^{-\frac{1}{2}}. \qquad (3.46)$$

The determinant over an infinite dimensional representation is somewhat formal; we can more concretely define it through its logarithm: $\log \mathrm{Det} = \mathrm{Tr} \log$. At this point we could represent the logarithm with a Schwinger parameterization (with an appropriate $i\epsilon$ prescription) to reproduce the integral form of the Wilson spool à la [7]. More simply, we can Taylor expand about the identity to recover

$$\log Z_{\Delta,\mathsf{s}}^{\mathrm{TAdS}_3} = \sum_{\mathcal{R}_{\Delta,\mathsf{s}}^{\mathrm{LW}}} \sum_{n=1}^{\infty} \frac{1}{n} \mathrm{Tr}_{\mathsf{R}_L} \left[ \mathcal{P} \exp \left( n \oint_{\gamma} A_L \right) \right] \mathrm{Tr}_{\mathsf{R}_R} \left[ \mathcal{P} \exp \left( -n \oint_{\gamma} A_R \right) \right]. \qquad (3.47)$$

Here the representations $\mathsf{R}_L \otimes \mathsf{R}_R$ are again taken to lie in $\mathcal{R}_{\Delta,\mathsf{s}}^{\mathrm{LW}}$ when the geodesic length is taken by convention to be positive.[21] Upon observing that the conjugacy classes of Thermal AdS are simply $\gamma_0^n$, $n \in \mathbb{Z}$ this restores (3.2).

We can now ask why this procedure might be more challenging to apply in non-elementary quotients. The key is actually in **Condition II**. While thermal AdS$_3$ is a quotient of $\mathbb{H}^3$, the boundary of thermal AdS$_3$ is not a quotient of $\partial\mathbb{H}^3 \cong \mathbb{CP}^1$. There are two limit points of the quotient, (without loss of generality) the points $(0, \infty) \in \mathbb{CP}^1$, which are not identified with any point on the boundary of thermal AdS$_3$. As such, eigenfunctions which are well behaved on the boundary of thermal AdS$_3$ need not lift to eigenfunctions on $\mathbb{H}^3$ which have a well defined limit approaching $(0, \infty)$. Indeed this is the case for the eigenfunctions associated to representations $\mathcal{R}_{\Delta,\mathsf{s}}^{\mathrm{LW/HW}}$.

Once we turn to non-elementary quotients this story becomes more complicated; see Appendix B. The limit set of $\Gamma$ is typically a fractal subset, $\Lambda$, of the Riemann sphere, and the boundary of $\mathbb{H}^3/\Gamma$ is the quotient

$$\partial(\mathbb{H}^3/\Gamma) = (\partial\mathbb{H}^3 \setminus \Lambda)/\Gamma. \qquad (3.48)$$

---

[20]Up to an unfixed holomorphic function of $\Delta$.

[21]There is a slight sleight-of-hand in going from (3.46) to (3.47). Both highest- and lowest-weight representations contribute poles to (3.46), however they contribute identical poles. Thus in (3.47) we have chosen to double the lowest-weight contribution. Upon Schwinger parameterization $\mathcal{R}_{\Delta,\mathsf{s}}^{\mathrm{HW/LW}}$ obtain opposite $i\epsilon$ prescriptions which cause $\mathcal{R}_{\Delta,\mathsf{s}}^{\mathrm{HW}}$ to couple to negative length geodesics; see [7] for details.

Since boundary points contained in $\Lambda$ do not descend to boundary points in the quotient, the set of eigenfunctions which are regular on the quotient and its boundary is substantially larger than on thermal AdS$_3$. When viewed as eigenfunctions on $\mathbb{H}^3$, they are allowed to be badly behaved near the boundary when approaching points in the limit set $\Lambda$. It is thus not enough simply to consider the set of irreducible representations $\mathcal{R}^{\mathrm{LW/HW}}_{\Delta,\mathsf{s}}$ to describe them. Furthermore $\mathbb{H}^3/\Gamma$ has multiple non-trivial cycles and thus we need to impose holonomy conditions around multiple cycles simultaneously while maintaining the correct multiplicity of poles in $Z^2_{\Delta,\mathsf{s}}$.

In fact our previous explorations into the worldline formalism provide the avenue for addressing both of these issues: we can decompose $\Gamma$ into centralizer subgroups, which for loxodromic or hyperbolic elements contain two fixed points. Within each subgroup, this reduces the DHS problem again to that on a torus with just one cycle. We can now utilize the same set of representations, $\mathcal{R}^{\mathrm{LW/HW}}_{\Delta,\mathsf{s}}$ for each centralizer subgroup. Thus when $\Gamma$ is purely loxodromic or hyperbolic we again write

$$Z^{\mathbb{H}^3/\Gamma}_{\Delta,\mathsf{s}} = \prod_{\mathcal{R}^{\mathrm{LW}}_{\Delta,\mathsf{s}}} \prod_{[\Gamma_0]_+} \mathrm{Det}_{\mathsf{R}_L \otimes \mathsf{R}_R} \left( 1 - \mathcal{P}\exp\left( \oint_{\gamma_0} A_L \right) \mathcal{P}\exp\left( -\oint_{\gamma_0} A_R \right) \right)^{-1}. \qquad (3.49)$$

Taking the log and performing the Taylor expansion then leads our result for the Wilson spool:

$$\log Z^{\mathbb{H}^3/\Gamma}_{\Delta,\mathsf{s}} = \sum_{\mathcal{R}^{\mathrm{LW}}_{\Delta,\mathsf{s}}} \sum_{[\Gamma_0]_+} \sum_{n=1}^{\infty} \frac{1}{n} \mathrm{Tr}_{\mathsf{R}_L}\left[ \mathcal{P}\exp\left( n\oint_{\gamma_0} A_L \right) \right] \mathrm{Tr}_{\mathsf{R}_R}\left[ \mathcal{P}\exp\left( -n\oint_{\gamma_0} A_R \right) \right]. \qquad (3.50)$$

Note that in the above formula we are summing over *primitive* conjugacy classes with positive geodesic length. Remarkably the determinant formula, (3.49), takes a form that is very reminiscent to a Selberg zeta function. Indeed given the matching established between holonomies of $A_{L/R}$ and conjugacy classes of $\gamma_0$ as group elements, (3.49) takes the form of a product of descendant weights under the action of primitive generators of $\Gamma$; we encourage the reader to compare to equation (4.2) of [29] or, more modernly (and suggestively), equation (3.2) of [30]. We will not make the connection between the Wilson spool and the Selberg zeta function explicit in this paper, however we are aware of upcoming work of other authors in this direction [15].

## 4 Discussion

In this paper we have extended the 'Wilson spool' prescription for coupling massive fields to three-dimensional gravity to any smooth cusp-free hyperbolic three-manifold. The result is the expression of the one-loop determinant as a topological line operator that wraps all possible non-trivial cycles of the background topology. Our construction follows from the realization of such manifolds as quotients of $\mathbb{H}^3$ by a discrete torsion-free subgroup

of isometries, $\Gamma$, and accommodating the structure of the quotient into the spool prescription. We provided three separate constructions of the spool: from the Selberg trace formula, from the physically intuitive worldline perspective, and from the quasinormal mode method which puts in the context in which the original spool results were derived. Our construction reproduces known results in the literature when available, and extends them to any massive spinning field on a smooth cusp-free hyperbolic manifold. There are several open question and future directions which we discuss below.

**Orbifolds, cusps and Lens spaces**  Beyond the spaces considered in this paper it would be interesting to understand how to construct the spool on a more general set of hyperbolic geometries, such as orbifolds and cusped spaces. When constructable as a quotient of $\mathbb{H}^3$ these arise if $\Gamma$ contains elliptic or parabolic elements respectively. The Selberg trace formula has been extended to finite volume quotients containing cusps and orbifold singularities [25], however the result does not have such a clear breakdown into characters of $\mathrm{SL}(2,\mathbb{R})$.

In the case of cusps the approach is very unclear; closed curves associated to a parabolic element are not homotopic to any geodesic in the bulk geometry. While one can attempt to consider a 'geodesic at infinity' as the curve is pushed towards the boundary, such a geodesic has both zero length and torsion rendering the corresponding character divergent. This is consistent with (3.4) and (3.5) from which we might consider directly the trace of a parabolic element within the appropriate highest/lowest weight representations

$$\mathrm{Tr}_{\mathsf{R}_j^{\mathrm{HW/LW}}} e^{zL_+} = \sum_{p=0}^{\infty} \langle j,p|e^{zL_+}|j,p\rangle_{\mathrm{HW/LW}} \to \infty \; . \tag{4.1}$$

We can understand some of the technical difficulties associated to elliptic elements more clearly by considering the quasinormal mode approach. Following the procedure of Section 3.2.2 the one-loop determinant on $\mathbb{H}^3/\Gamma$ can still be reduced to a sum over quotients of $\mathbb{H}^3$ along a single axis. However, when the centralizer also contains elliptic elements as described in (B.11) this leads to a couple of extra subtleties. Firstly, we now must consider *two* separate cycles in the geometry and as yet we do not have a construction to deal with this; see Figure 5. Secondly, the geodesics associated to elliptic elements are zero length, and as such the characters

$$\chi_j^{\mathrm{HW/LW}}(\alpha\tau) \tag{4.2}$$

have poles on the real $\alpha$-axis. The more rigorous derivation of the spool in the quasinormal mode formalism [7] suggests that when this occurs the Wilson spool cannot be written as a discrete sum but must instead be taken as a contour integral which picks up residues due to the poles of the character.

A greater understanding of these issues should also provide insight into the de Sitter spool on Lens spaces. In this case also there are multiple holonomy conditions to incorporate into the quasinormal mode method. Furthermore in de Sitter the associated characters which appear also involve poles on the real $\alpha$-axis of a contour integral, although the associated geodesics are not zero length.

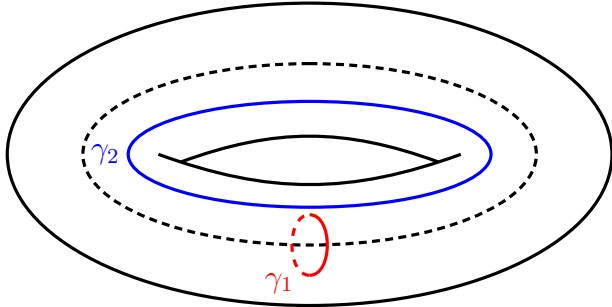

**Figure 5:** The simplest example of an elliptic element taken along the same axis as a hyperbolic element resulting in a thermal AdS$_3$ with a conical defect (depicted as the dashed line) as its quotient. In this case there are two nontrivial cycles in the geometry.

**On the integral form of the spool**  The above discussion point highlights the following comment made in the Section 1: we have primarily presented the results in this paper as a sum over windings on a set of primitive free-loops, (3.2). This makes manifest its 'spooling' nature and its connection to worldline quantum mechanics. Alternatively, a simple rewriting casts this sum as contour integral, e.g. (3.3). This seems like mostly an aesthetic choice in this work, however the trivial relation between sum and the integral might be an artifact of the torsion-free quotients we have considered: every centralizer is an infinite cyclic group and so has a natural structure of wrapping a primitive element arbitrarily many times.

More generally we expect the integral form of the Wilson spool to be the more comprehensive description. This was already evident in the original works of [5,6] for one-loop determinants in de Sitter spacetimes where the contour integral picks up additional poles and reproduces the intricate meromorphic structure of the one-loop determinant. Additionally for the Wilson spool applied to JT gravity [8] the integral expression is unavoidable: the contour defining $\mathbb{W}$ is open and while certain scenarios allow portions of the contour to wrap poles that give $\mathbb{W}$ a spooling interpretation, there will remain an open segment. This segment is important for reproducing universal logarithmic divergences that appear in two dimensions.

As emphasized in the previous point, a better understanding of the interplay between integral contour prescriptions, the Selberg trace formula, and the incorporation of simultaneous conditions in quasinormal modes will likely be key to a more comprehensive construction of $\mathbb{W}$ that includes torsion quotients of $\mathbb{H}^3$ and $S^3$.

**Adding matter to the Virasoro TQFT**  The primary results of the present work rely on the classical equivalence between 3D gravity with negative cosmological constant and two copies of $\mathrm{SL}(2, \mathbb{R})$ quantum gravity. While this is sufficient for the semi-classical computations and perturbatively off-shell statement claims of this paper, a full quantum treatment of 3D gravity coupled to matter requires a more careful treatment. As mentioned in the introduction, pure 3D gravity is *not* equivalent to Chern-Simons theory at

the quantum level, since not every flat connection corresponds to an invertible metric. However, a careful treatment of the difference between gravity and Chern-Simons theory allows one to define a topological field theory for the former, known as the *Virasoro TQFT* [16,17]. Like Chern-Simons theory, the Virasoro TQFT admits extended operators analogous to Wilson lines. A promising route, then, for a fully quantum treatment of 3D gravity coupled to matter may be found in embedding the Wilson spool in the Virasoro TQFT. This is currently under investigation by the present authors.

## Acknowledgements

We thank Alejandra Castro, Samuel Haupfear, Albert Law, Victoria Martin, Andy Svesko, and Claire Zukowski for helpful discussions and additionally to Alejandra Castro and Claire Zukowski for comments on a draft of this paper. We especially thank the authors of [15] for discussions on their upcoming work. This work has been partially supported by STFC consolidated grants ST/T000694/1 and ST/X000664/1. JRF is additionally supported by Simons Foundation Award number 620869.

## A $\mathfrak{sl}(2, \mathbb{R})$ conventions

The $\mathfrak{sl}(2, \mathbb{R})$ algebra can be written in a ladder operator basis $\{L_0, L_\pm\}$ satisfying

$$[L_\pm, L_0] = \pm L_\pm , \qquad [L_+, L_-] = 2L_0 . \tag{A.1}$$

The quadratic Casimir of $\mathfrak{sl}(2, \mathbb{R})$ in this basis is given by

$$c_2^{\mathfrak{sl}}(2, \mathbb{R}) = L_0^2 - \frac{1}{2}(L_- L_+ + L_+ L_-) , \tag{A.2}$$

and is a constant on irreducible representations of $\mathfrak{sl}(2, \mathbb{R})$. The representation theory of $\mathfrak{sl}(2, \mathbb{R})$ is rich[22] however in this paper we will focus on lowest- and highest-weight representations which we denote as $\mathsf{R}_j^{\mathrm{LW/HW}}$, respectively.

Lowest-weight representations are built from a lowest-weight state satisfying

$$L_0|j, 0\rangle_{\mathrm{LW}} = j|j, 0\rangle_{\mathrm{LW}} , \qquad L_+|j, 0\rangle_{\mathrm{LW}} = 0 , \tag{A.3}$$

and acting with powers of $L_-$ which raises the weight:

$$|j, p\rangle_{\mathrm{LW}} = (L_-)^p |j, 0\rangle_{\mathrm{LW}} , \qquad L_0|j, p\rangle_{\mathrm{LW}} = (j + p)|j, p\rangle_{\mathrm{LW}} . \tag{A.4}$$

Similarly, highest-weight representations are built from a highest-weight state satisfying

$$L_0|j, 0\rangle_{\mathrm{HW}} = -j|j, 0\rangle_{\mathrm{HW}} , \qquad L_-|j, 0\rangle_{\mathrm{HW}} = 0 , \tag{A.5}$$

---

[22]See [31] for a comprehensive summary.

and acting with powers of $L_+$ which lowers the weight:

$$|j, p\rangle_{\text{HW}} = (L_+)^p \, |j, 0\rangle_{\text{HW}} \, , \qquad L_0|j, p\rangle_{\text{HW}} = (j + p)|j, p\rangle_{\text{HW}} \, . \tag{A.6}$$

Our conventions are such that for both representations the quadratic Casimir is given by

$$c_2^{\mathfrak{sl}(2,\mathbb{R})}|j, 0\rangle_{\text{LW/HW}} = j(j - 1)|j, 0\rangle_{\text{LW/HW}} \, . \tag{A.7}$$

The characters of highest- and lowest-weight representations are defined as

$$\chi_j^{\text{LW/HW}}(\tau) \equiv \text{Tr}_{\text{R}_j^{\text{LW/HW}}} \left( q^{L_0} \right) \, , \qquad q = e^{2\pi i \tau} \, . \tag{A.8}$$

Plugging in the $L_0$ spectrum in the LW/HW representations, the characters are given by simple geometric series:

$$\chi_j^{\text{LW}}(\tau) = \sum_{p=0}^{\infty} q^{j+p} = \frac{q^j}{1 - q} = \frac{e^{i\pi\tau(2j-1)}}{2\sinh(-i\pi\tau)} \, ,$$

$$\chi_j^{\text{HW}}(\tau) = \sum_{p=0}^{\infty} q^{-j-p} = \frac{q^{-j}}{1 - q^{-1}} = \frac{e^{-i\pi\tau(2j-1)}}{2\sinh(i\pi\tau)} \, , \tag{A.9}$$

with the sums converging for $|q| < 1$ in the case of $\chi_j^{\text{LW}}(\tau)$ and $|q| > 1$ in the case of $\chi_j^{\text{HW}}(\tau)$.

# B   Details on hyperbolic quotients

In this appendix we review some basic facts about hyperbolic three-manifolds used in the main text.

The most basic hyperbolic three-manifold is global hyperbolic three-space, $\mathbb{H}^3$, which can be modeled by the Poincaré ball embedded in $\mathbb{R}^3$ with metric

$$ds_{\mathbb{H}^3}^2 = \frac{\mathrm{d}x_i \mathrm{d}x^i}{(1 - |x|^2)^2} \, . \tag{B.1}$$

The conformal boundary of $\mathbb{H}^3$ is the two-sphere of points $|x|^2 = 1$, which we will often identify with the extended complex plane. The group of isometries of $\mathbb{H}^3$ is $\text{SO}(3, 1) \cong \text{PSL}(2, \mathbb{C})$.

Locally, every hyperbolic three-manifold is isometric to $\mathbb{H}^3$, i.e. its metric tensor can always be brought into the form (B.1) using an appropriate change of coordinates. Globally, every hyperbolic three-manifold can be expressed as a quotient space $\mathbb{H}^3/\Gamma$ where $\Gamma \subset \text{PSL}(2, \mathbb{C})$ is some discrete subgroups of the isometry group of $\mathbb{H}^3$. Discrete subgroups of $\text{PSL}(2, \mathbb{C})$ are known as Kleinian groups, and the study of hyperbolic three-manifolds is equivalent to the study of Kleinian groups.

Elements $\gamma \in \text{PSL}(2, \mathbb{C})$ come in four basic types, depending on the value of $\text{tr}(\gamma)$ in the fundamental representation:

- If $\mathrm{tr}(\gamma)^2 = 4$, then $\gamma$ is **parabolic**. All parabolic elements are conjugate to

$$\begin{pmatrix} 1 & 1 \\ 0 & 1 \end{pmatrix}. \tag{B.2}$$

Parabolic elements of $\mathrm{PSL}(2,\mathbb{C})$ act freely on $\mathbb{H}^3$, but leave fixed a single point on the boundary sphere.

- If $0 \le \mathrm{tr}(\gamma)^2 < 4$, then $\gamma$ is **elliptic**. All elliptic generators are conjugate to the matrix

$$\begin{pmatrix} e^{i\theta} & 0 \\ 0 & e^{-i\theta} \end{pmatrix}, \tag{B.3}$$

for some real angle $\theta$. The fixed-point set of an elliptic element is a geodesic in $\mathbb{H}^3$.

- If $\mathrm{tr}(\gamma)^2 \ge 4$, then $\gamma$ is **hyperbolic**. All hyperbolic elements are conjugate to

$$\begin{pmatrix} \lambda & 0 \\ 0 & \lambda^{-1} \end{pmatrix} \tag{B.4}$$

for some real $\lambda$. Hyperbolic elements act freely on $\mathbb{H}^3$, and leave fixed (as a set) a single geodesic.

- Finally, if $\mathrm{tr}(\gamma)$ is not real, then $\gamma$ is said to be **loxodromic**. All loxodromic elements are conjugate to

$$\begin{pmatrix} q^{1/2} & 0 \\ 0 & q^{-1/2} \end{pmatrix} \tag{B.5}$$

with $q$ not real and $|q| \ne 1$. Defining $q = e^{2\pi i\tau}$, then

$$\mathrm{tr}(\gamma) = 2\cos(\pi\tau). \tag{B.6}$$

Similarly to hyperbolic transformations, a loxodromic element of $\mathrm{PSL}(2,\mathbb{C})$ acts freely on $\mathbb{H}^3$, but fixes a preferred geodesic.

In practice, it is useful to group together hyperbolic and loxodromic elements by allowing $q$ to be real (equivalently, allowing $\tau$ to be pure imaginary). The quotient space $\mathbb{H}^3/\Gamma$ is smooth if and only if $\Gamma$ contains only hyperbolic and loxodromic elements. In this case we say that $\Gamma$ is 'torsion-free.' If $\Gamma$ contains an elliptic generator, then the quotient $\mathbb{H}^3/\Gamma$ will have a conical singularity of deficit angle $\theta$ along the corresponding fixed geodesic. If $\Gamma$ contains a parabolic subgroup, then $\mathbb{H}^3/\Gamma$ will have a 'cusp'.

**The geometry of the quotient:** Let us now consider the case in which $\Gamma$ contains only hyperbolic and loxodromic elements, so that the quotient $M = \mathbb{H}^3/\Gamma$ is smooth. Since $M$ is a smooth three-manifold whose universal covering space is $\mathbb{H}^3$, then there is a natural isomorphism

$$\pi_1(M, p) \cong \Gamma, \tag{B.7}$$

for each basepoint $p$. Essentially, each closed loop in $M$ can be lifted to an open loop in $\mathbb{H}^3$ which connects the point $p$ and $\gamma \cdot p$ for some element $\gamma \in \Gamma$. Indeed, the topology of $M$ is completely dictated by the Kleinian group $\Gamma$.

The elements of $\Gamma$ are also in one-to-one correspondence with closed geodesics in $M$, based at $p$. This essentially boils down to the fact that the hyperbolic metric on $M$ is induced by the hyperbolic metric on $\mathbb{H}^3$, and that on $\mathbb{H}^3$ there is a unique geodesic connecting the points $p$ and $\gamma \cdot p$. If we do not care about the base point, then there is a one-to-one correspondence between conjugacy classes of $\Gamma$ and homotopy classes of geodesics on $M$.

Finally, the boundary of $M$ is determined by the action of $\Gamma$ on the boundary $\mathbb{CP}^1 \cong \partial \mathbb{H}^3$ of hyperbolic 3-space. However, there are some subtle points in this construction. The group $\Gamma$ acts on the boundary sphere via Möbius transformations, but the action of $\Gamma$ may contain limit points where $\Gamma$ does not act properly discontinuously. The set of such limit points, usually denoted by $\Lambda$, is typically a very complicated fractal object, and must be removed from the boundary in order to define a good quotient. With this in mind, the boundary of $M$ is the quotient of $\mathbb{CP}^1 \setminus \Lambda$ by the action of $\Gamma$. This construction also endows the boundary of $M$ with a natural complex structure induced from the complex structure on $\mathbb{CP}^1$.

**Thermal AdS:** As a simple example to make the above discussion more concrete, take $\Gamma \cong \mathbb{Z}$ to be generated by a single element

$$\gamma = \begin{pmatrix} q^{1/2} & 0 \\ 0 & q^{-1/2} \end{pmatrix}, \tag{B.8}$$

for some complex number $q$ not on the unit circle. In this case, the quotient $M = \mathbb{H}^3/\Gamma$ is a solid torus, i.e. thermal AdS$_3$. Since the solid torus has one non-contractible loop, we clearly have

$$\pi_1(M) \cong \mathbb{Z} \cong \Gamma. \tag{B.9}$$

Furthermore, homotopy classes of closed geodesics on $M$ are labeled by their winding number, and so are in one-to-one correspondence with elements of $\Gamma$.[23]

The action of $\Gamma$ on the boundary is straightforward. Let $z$ be an inhomogeneous coordinate on $\mathbb{CP}^1$. Then $\gamma$ acts on $\mathbb{CP}^1$ by sending $z$ to $qz$. There are two points on the sphere, $z = 0$ and $z = \infty$, which are completely fixed by $\Gamma$. These turn out to be the only limit points, i.e. $\Lambda = \{0, \infty\}$. By the discussion above, the asymptotic boundary of $M$ is

$$\partial M = (\mathbb{CP}^1 \setminus \Lambda)/\Gamma = (\mathbb{C} \setminus \{0\})/(z \sim qz), \tag{B.10}$$

which is a torus. This is most easily seen by setting $q = e^{2\pi i \tau}$ and making the coordinate transformation $z = e^{2\pi i u}$, so that $u \sim u + 1 \sim u + \tau$. The modular parameter $\tau$ then labels the complex structure on $\partial M$.

---

[23]Since $\Gamma$ is Abelian, the set of conjugacy classes in $\Gamma$ is just $\Gamma$ itself.

**Multiplicity of loxodromic elements:** In the main text, we find it useful to introduce the notion of the multiplicity of a loxodromic (or hyperbolic) element $\gamma \in \Gamma$. Given such an element, we can construct $\mathcal{C}(\gamma)$, its centralizer in $\Gamma$, which must take the form [25]

$$\mathcal{C}(\gamma) = \langle \gamma_0 \rangle \times E(\gamma) \tag{B.11}$$

where $\langle \gamma_0 \rangle$ is an infinite cyclic group generated by an element $\gamma_0 \in \mathcal{C}(\gamma)$ and $E(\gamma)$ is the subgroup of finite order elements within $\mathcal{C}(\gamma)$. We call $\gamma_0$ a primitive element for $\gamma$ in $\Gamma$. Within this decomposition we can write $\gamma = \gamma_0^n e$ for some $e \in E(\gamma)$. While the decomposition (B.11) is not unique, the 'multiplicity'

$$n_\gamma = |n| \tag{B.12}$$

does not depend on the choice we make. Furthermore $n_\gamma$ is also an invariant of the conjugacy class $[\gamma]$ in $\Gamma$, independent of the choice of representative.

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
