# Peer review of "A spool for every quotient: One-loop partition functions in AdS$_3$ gravity"

_SciPost Physics_

## Round 1 · Referee Report · Anonymous (Referee 1) · 2025-11-7

Report

The manuscript proposes an extension of the “Wilson spool” framework for coupling free massive fields to $AdS_3$ gravity. The authors generalize the construction, previously developed for Euclidean $dS_3$ and rotating BTZ black holes, to all smooth, cusp-free hyperbolic 3-manifolds. They verify the prescription by reproducing known results for massive scalars and vectors on such spaces. One highlight is that this prescription enables the authors to extend these results to massive fields of arbitrary integer spin. The authors also motivate their prescription from the perspectives of the Selberg trace formula, the worldline path integral, and the quasinormal mode method.

The manuscript is well written and the analysis is clearly presented. While the advance is mainly technical, the construction is useful for researchers working on $AdS_3$ gravity. I recommend publication once the authors address the following minor points:

  1. In the spool expressions (1.3) and (1.4), are the relevant representations and connections those of $SL(2,\mathbb R)$? If so, please clarify why this choice is appropriate in the Euclidean setting, where $PSL(2,\mathbb C)$ (or $SO(1,3)$) would appear to be the natural isometry group.

  2. Just before equation (2.9), the symbol $\tau$ seems to be a typo and should likely read $\vartheta$.

  3. In Footnote 12, it might be worth noting that obtaining the full one-loop partition function for a massless spin-2 field requires dividing by the corresponding ghost determinant, in addition to taking the massless limit of the massive spin-2 result.

Recommendation

Ask for minor revision

---

## Round 1 · Referee Report · Anonymous (Referee 2) · 2025-11-28

Strengths

  1. The generalization of the `Wilson spool' approach of writing partition functions in 3 d gravity to arbitrary massive spin fields in all smooth cusp-free solutions of Euclidean gravity with negative cosmological constant.

  2. There is a test by comparting the resulting expression (eq 3.3) against the Selberg Trace formula for fields of spin=0, 1, 2,

Weaknesses

  1. There is no test of the in formula in 3.3 fo spins >2 using another approach say the Selberg trace formula. For the case of the BTZ case or the solid torus, the expression for the one loop determinant from the Wilson spool approach was tested against the Selberg trace for arbitrary massive spinning fields in https://arxiv.org/pdf/2507.05358.

It would have been good if there had been some tests for at least some other quotient of H_3.

  1. The general program of the Wilson spool approach in 3 d gravity gives empahsis to the Chern-Simons formulation over the metric formulation. For fields, massive or massless in 3d, one standard quantity which is evaluated in the conventional metric formalism are bulk 2 point functions or in the case of $AdS_3$ bulk boundary propagators. It would be worthwhile if the authors can comment on how these can be reformulated in the Wilson-spool formulation. Do, they correspond to expressions similar to 3.3, but with open Wilson lines.

The authors can just point that such questions do remain open in this effort at reformulating 3d gravity including external (massive and massless/spinning) fields in the Wilson-spool approach.

Report

The paper generalises the program ref 6, 5 and followed up in 7 to writing
one loop partition functions of massive spinning fields in 3 d gravity on smooth
cusp free solutions of 3d gravity. The earlier work in particular ref 7 carries out the analysis for dS_3 and AdS_3.

The paper is based on sound principles, the final result 3.3, is a natural
generalisation of earlier expressions say 1.4 of ref 7, to eq 3.3, in which there is a sum over unoriented non-contractable loops.

The generalisation is expected, it would have been more useful if the authors carried out some non-trivial tests for spins>2

Requested changes

I request the authors to comment on the use of the Wilson spool approach
to evaluate 2 point function either bulk 2 point functions or bulk boundary propagators of fields (even scalars)
in 3 d gravity.
It is important to point out the Wilson spool approach needs to be generalised to this cases and possibly can be generalised.

Also, I request the authors to comment on the possibility of using this
approach to evaluate the one loop corrections to entanglement entropy
on the lines of https://arxiv.org/pdf/1306.4682.
Note that the Selberg trace formula was used in eq 36 of this paper,here quaotients of H_3 involved the Schottky group.
Is it possibile to generalise the Wilson loop formualism to such quotients ?

There there would be a possibility of testing the expressions even for massive spining fields of s>2.

Recommendation

Ask for minor revision

---

## Editorial Decision

awaiting_resubmission